# GENERALIZATION BOUNDS AND MODEL COMPLEXITY FOR KOLMOGOROV–ARNOLD NETWORKS

**Xianyang Zhang**
Department of Statistics
Texas A&M University
College Station, TX 77843, USA
zhangxiany@stat.tamu.edu

**Huijuan Zhou**
School of Statistics and Data Science
Shanghai University of Finance and Economics
Shanghai, China
zhouhuijuan@mail.shufe.edu.cn

## ABSTRACT

Kolmogorov–Arnold Network (KAN) is a network structure recently proposed in Liu et al. (2024c) that offers improved interpretability and a more parsimonious design in many science-oriented tasks compared to multi-layer perceptrons. This work provides a rigorous theoretical analysis of KAN by establishing generalization bounds for KAN equipped with activation functions that are either represented by linear combinations of basis functions or lying in a low-rank Reproducing Kernel Hilbert Space (RKHS). In the first case, the generalization bound accommodates various choices of basis functions in forming the activation functions in each layer of KAN and is adapted to different operator norms at each layer. For a particular choice of operator norms, the bound scales with the $l_1$ norm of the coefficient matrices and the Lipschitz constants for the activation functions, and it has no dependence on combinatorial parameters (e.g., number of nodes) outside of logarithmic factors. Moreover, our result does not require the boundedness assumption on the loss function and, hence, is applicable to a general class of regression-type loss functions. In the low-rank case, the generalization bound scales polynomially with the underlying ranks as well as the Lipschitz constants of the activation functions in each layer. These bounds are empirically investigated for KANs trained with stochastic gradient descent on simulated and real data sets. The numerical results demonstrate the practical relevance of these bounds.

## 1 INTRODUCTION

The Kolmogorov-Arnold representation theorem (KART) states that if $f$ is a multivariate continuous function defined on $[0, 1]^d$, then $f$ can be written as a finite composition of continuous functions of a single variable, and the binary operation of addition. More specifically, $f$ can be represented by a two-layer network in the form of

$$f(x_1, \ldots, x_d) = \sum_{i=0}^{2d} \psi_i \left( \sum_{j=1}^{d} \psi_{i,j}(x_j) \right),$$

where $\psi_{i,j} : [0, 1] \to \mathbb{R}$ and $\psi_i : \mathbb{R} \to \mathbb{R}$. Motivated by KART, Liu et al. (2024c) introduced the Kolmogorov–Arnold Networks (KANs), expanding on the original two-layer network to accommodate arbitrary depths. Mathematically, the KANs can be concisely described through the compositions of $L$ multivariate vector-valued functions:

$$\text{KAN}(\mathbf{x}) = \mathbf{\Psi}_L \circ \mathbf{\Psi}_{L-1} \circ \cdots \circ \mathbf{\Psi}_1(\mathbf{x}), \quad \mathbf{x} \in \mathbb{R}^{d_0},$$

where $\mathbf{\Psi}_i$ is a matrix of univariate functions:

$$\mathbf{\Psi}_i = \begin{pmatrix} \psi_{i,1,1} & \psi_{i,1,2} & \cdots & \psi_{i,1,d_{i-1}} \\ \psi_{i,2,1} & \psi_{i,2,2} & \cdots & \psi_{i,2,d_{i-1}} \\ \vdots & \vdots & \ddots & \vdots \\ \psi_{i,d_i,1} & \psi_{i,d_i,2} & \cdots & \psi_{i,d_i,d_{i-1}} \end{pmatrix}$$

with $\psi_{i,j,k} : \mathbb{R} \to \mathbb{R}$. Here, we have defined

$$\mathbf{\Psi}_i(\mathbf{x}) = \left( \sum_{j=1}^{d_{i-1}} \psi_{i,1,j}(x_j), \dots, \sum_{j=1}^{d_{i-1}} \psi_{i,d_i,j}(x_j) \right)^\top \tag{1}$$

for $\mathbf{x} = (x_1, \dots, x_{d_{i-1}})^\top \in \mathbb{R}^{d_{i-1}}$ and $1 \leq i \leq L$. The node values for the $i$th layer are given by $\mathbf{x}_i = (x_{i,1}, \dots, x_{i,d_i})^\top = \mathbf{\Psi}_i \circ \mathbf{\Psi}_{i-1} \circ \cdots \circ \mathbf{\Psi}_1(\mathbf{x})$ where $x_{i,j} = \sum_{k=1}^{d_{i-1}} \psi_{i,j,k}(x_{i-1,k})$ is defined recursively.

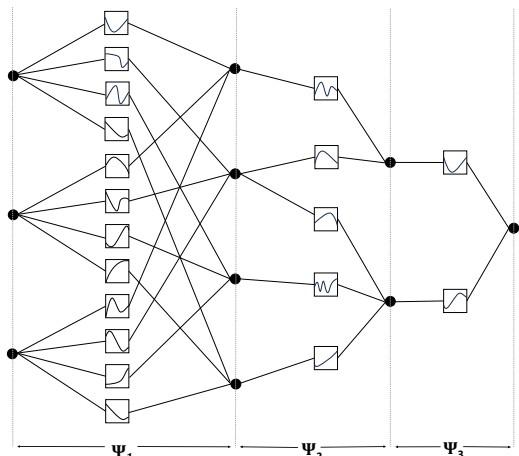

Figure 1: Illustration of Kolmogorov-Arnold Networks, where the edges are associated with one-dimensional trainable functions while the nodes perform summation.

In contrast to multi-layer perceptions (MLPs), which use fixed activation functions on nodes, KANs utilize learnable activation functions along the edges. This architectural shift eliminates the need for conventional linear weight matrices. Instead, KANs replace each weight parameter with a trainable one-dimensional function. In KANs, nodes serve as aggregation points, summing incoming signals without applying non-linear transformations; see Figure 1 for an illustration. Liu et al. (2024c) demonstrated the potential of KANs for science-oriented tasks due to its accuracy and interpretability. Subsequent research has explored the application of KANs in various domains such as graphs (Bresson et al., 2024; De Carlo et al., 2024; Kiamari et al., 2024), partial differential equations (Wang et al., 2024; Shukla et al., 2024), operator learning (Abueidda et al., 2024; Nehma & Tiwari, 2024), tabular data (Poeta et al., 2024), time series (Genet & Inzirillo, 2024; Vaca-Rubio et al., 2024; Xu et al., 2024b), human activity recognition (Liu et al., 2024b), neuroscience (Herbozo Contreras et al., 2024; Yang et al., 2024), quantum science (Kundu et al., 2024; Li et al., 2024b), computer vision (Azam & Akhtar, 2024; Bodner et al., 2024; Cheon, 2024; Li et al., 2024a; Seydi, 2024b), kernel learning (Zinage et al., 2024), nuclear physics (Liu et al., 2024a), electrical engineering (Peng et al., 2024), and biology (Pratyush et al., 2024). Unlike MLPs, KANs allow a flexible way to specify and estimate the activation functions from training samples. Various choices of activation functions have been recently investigated, including B-splines Liu et al. (2024c), wavelets (Seydi, 2024b; Bozorgasl & Chen, 2024), radial basis functions (Li, 2024; Ta, 2024), Fourier series (Xu et al., 2024a), finite basis (Howard et al., 2024), Jacobi basis functions (Aghaei, 2024a), polynomial basis functions (Seydi, 2024a), and rational functions (Aghaei, 2024b). Additional techniques for KANs have also been proposed, such as regularization (Altarabichi, 2024), Kansformer (combining transformer with KANs) (Chen et al., 2024), adaptive grid update (Rigas et al., 2024), federated learning (Zeydan et al., 2024), and Convolutional KANs (Bodner et al., 2024). A survey on the recent development of KANs can be found in Hou & Zhang (2024).

## 1.1 CONTRIBUTIONS

Despite the increasing interest in applying KANs to various scientific problems and exploring the best ways to construct the network structure and training process, there is currently no recent research focusing on quantifying the complexity of KANs to the best of our knowledge. Additionally, there is a need to understand how the performance of KANs is impacted by network structure and its complexity. The contribution of this work can be summarized as follows.

- Theorem 3 below will give the rigorous statement of the generalization bound for KANs, which (i) accommodates different choices of basis functions in forming the activation functions in each layer; (ii) is adapted to various operator norms at each layer; (iii) scales with the $l_1$ norm of the coefficient matrices and the Lipschitz constants for the activation functions for a particular choice of operator norms; (iv) has no dependence on combinatorial parameters (e.g., number of nodes) outside of logarithmic factors; (v) does not require the boundedness assumption on the loss function.

- Theorem 4 presents the generalization bound when the activation functions for each layer belong to a Reproducing Kernel Hilbert Space (RKHS) with a low-rank structure. The bound scales polynomially with the underlying ranks as well as the Lipschitz constants of the activation functions in each layer.

- We empirically investigate these bounds when the KANs are trained with stochastic gradient descent (SGD) on simulated and real data sets. The results shed new light on how the performance of KANs is affected by the underlying network structure and the network complexity. In particular, we observe that the complexity measure derived in Section 2 is tightly correlated with the excess loss, demonstrating the complexity measure's practical relevance. This theoretical-empirical connection suggests a potential strategy for regularization, where the complexity measure could be used as a regularizer in designing the network structure and during the training process, either explicitly or implicitly.

## 1.2 RELATED WORKS

Generalization bounds for MLPs have long been studied in the literature on learning theory; see, e.g., Bartlett (1996); Anthony et al. (1999). More recently, Bartlett et al. (2017) provided a margin bound and corresponding Rademacher analysis that can be adapted to various operator norms at each layer. The proof technique therein was inspired by the early result for two-layer networks in Bartlett (1996). Farrell et al. (2021) established non-asymptotic bounds for MLPs for a general class of nonparametric regression-type loss functions. Schmidt-Hieber (2020) showed that MLPs with a sparsely connected structure achieve the minimax rates of convergence (up to a logarithmic factor) under a general composition assumption on the regression function. The aforementioned work has demonstrated that combining empirical process tools with considerations specific to deep neural networks (DNNs) can yield meaningful results. However, the practical relevance of these results for guiding DNN construction and training requires further investigation. Additionally, alternative theoretical tools have been developed to analyze DNNs, including the neural tangent kernel Jacot et al. (2018) and random matrix theory Mei & Montanari (2022).

The contribution of the present work is to provide a generalization bound for the novel KAN network structure, which is fundamentally different from the traditional MLPs as demonstrated in Liu et al. (2024c;b). The current analysis uses covering numbers and is closely related to earlier covering number bounds in Anthony et al. (1999) and the recent results in Bartlett et al. (2017) for MLPs. The current proof also utilizes Maurey's sparsification lemma with a suitable adaption to the current setting. However, it is important to highlight some key differences with existing results for MLPs. Firstly, our results can be seen as a non-trivial extension of MLP results to a more flexible network structure specified through a more general choice of activation functions. Secondly, Bartlett et al. (2017) focuses on the margin-based multiclass generalization bound, which has a bounded loss function (more precisely, the ramp loss). In contrast, our results cover a wider range of loss functions (including the ramp loss for classification and a general class of regression-type loss functions), which are not required to be bounded. Thirdly, the results for KANs with a low-rank structure in Section 2.3 appear to be new, and we are not aware of comparable results for MLPs in the recent literature.

The rest of the article is organized as follows: We provide the generalization bounds for KANs equipped with activation functions that are either represented by a set of basis functions or lying in a low-rank space in Sections 2.2 and 2.3, respectively. Section 3 presents numerical results to support the theoretical findings. Section 4 concludes and discusses a few open problems for future research. The appendix contains all the proofs, additional discussions, and numerical results.

## 2 STATISTICAL ANALYSIS

We focus on the supervised learning task, which aims to predict a response using a set of covariates. Suppose we have $n$ independent observations $(\mathbf{x}_i, y_i) \in \mathcal{X} \times \mathcal{Y}$ with $1 \leq i \leq n$ generated from some distribution $P$, where $\mathcal{X} \subset \mathbb{R}^{d_0}$ and $\mathcal{Y} \subset \mathbb{R}^{d_L}$. Let $\mathcal{L}(\cdot, \cdot)$ be a loss function defined on $\mathcal{Y} \times \mathcal{Y}$. We consider the KAN-induced neural network class

$$\mathcal{M} = \{f(\cdot) = \mathbf{\Psi}_L \circ \mathbf{\Psi}_{L-1} \circ \cdots \circ \mathbf{\Psi}_1(\cdot) : \mathbf{\Psi}_l \in \mathcal{F}_l, l = 1, 2, \ldots, L\},$$

for some pre-specified function classes $\mathcal{F}_l$, where $\mathbf{\Psi} \in \mathcal{F}_l$ is a map from $\mathbb{R}^{d_{l-1}}$ to $\mathbb{R}^{d_l}$. Given any $f \in \mathcal{M}$, we define the generalization error (or risk) $R(f) = \mathbb{E}[\mathcal{L}(f(\mathbf{x}), y)]$ for $(\mathbf{x}, y) \sim P$. To find the optimal predictor within $\mathcal{M}$, we minimize the empirical risk function

$$\min_{f \in \mathcal{M}} \frac{1}{n} \sum_{i=1}^{n} \mathcal{L}(f(\mathbf{x}_i), y_i). \tag{2}$$

Our goal here is to establish a bound on $R(f)$ for every $f \in \mathcal{M}$. In particular, let $\hat{f}$ be the predictor resulting from an optimization scheme (e.g., SGD, Adam, or L-BFGS) that solves (2). Our result implies a bound on the generalization error $R(\hat{f})$.

### 2.1 NOTATION AND DEFINITIONS

We introduce the notion of proper covering numbers. For a set $\mathcal{U}$ equipped with some norm $\|\cdot\|$, we let $\mathcal{N}(\mathcal{U}, \epsilon, \|\cdot\|)$ be the smallest number of a subset $\mathcal{V}$ of $\mathcal{U}$ such that for any $u \in \mathcal{U}$, there exists a $v \in \mathcal{V}$ so that $\|v - u\| \leq \epsilon$. Let $\mathbf{X} = [\mathbf{x}_1, \ldots, \mathbf{x}_n]^\top \in \mathbb{R}^{n \times d}$ for $\mathbf{x}_i \in \mathbb{R}^d$ and $\mathcal{F}$ be a class of multivariate vector-valued functions mapping from $\mathbb{R}^d$ to $\mathbb{R}^m$. For any $\mathbf{\Psi} \in \mathcal{F}$, we write $\mathbf{\Psi}(\mathbf{X}) = [\mathbf{\Psi}(\mathbf{x}_1), \ldots, \mathbf{\Psi}(\mathbf{x}_n)]^\top \in \mathbb{R}^{n \times m}$. For a matrix $A = (a_{ij})$, we denote by $\|A\|_2 = \sqrt{\sum_{i,j} a_{ij}^2}$ its Frobenius norm.

### 2.2 RISK ANALYSIS: ACTIVATION FUNCTIONS REPRESENTED BY BASIS FUNCTIONS

Let $\mathcal{H}_i(\mathbf{X}) := \{\mathbf{\Psi}_i \circ \cdots \circ \mathbf{\Psi}_1 \circ \mathbf{\Psi}_0(\mathbf{X}) : \mathbf{\Psi}_l \in \mathcal{F}_l, l = 1, 2, \ldots, i\}$, where $\mathbf{\Psi}_0(\mathbf{X}) = \mathbf{X}$ is the identity map. Given $\epsilon_i, \rho_i > 0$ for $1 \leq i \leq L$, we define $s_1 = \epsilon_1$ and $s_k = \sum_{i=1}^{k} (\prod_{j=i+1}^{k} \rho_j) \epsilon_i$ for $k = 2, \ldots, L$, where we set $\prod_{j=a}^{b} \rho_j = 1$ for $a > b$. The subscript $i$ in the norm $|\cdot|_i$ below merely indicates an index and does not refer to any $l_i$ norm.

**Proposition 1.** *Suppose $\mathbf{\Psi}_i \in \mathcal{F}_i$ is a map from $\mathbb{R}^{d_{i-1}}$ to $\mathbb{R}^{d_i}$, which satisfies $|\mathbf{\Psi}(\mathbf{X}) - \mathbf{\Psi}(\mathbf{X}')|_i \leq \rho_i |\mathbf{X} - \mathbf{X}'|_{i-1}$ for $\rho_i > 0$ and $\mathbf{X}, \mathbf{X}' \in \mathbb{R}^{n \times d_{i-1}}$. Then we have*

$$\begin{aligned}
&\mathcal{N}(\mathcal{H}_L(\mathbf{X}), s_L, |\cdot|_L) \\
&\leq \prod_{i=0}^{L-1} \sup_{\mathbf{\Psi}_1, \ldots, \mathbf{\Psi}_i} \mathcal{N}(\{\mathbf{\Psi} \circ \mathbf{\Psi}_i \circ \mathbf{\Psi}_{i-1} \circ \cdots \circ \mathbf{\Psi}_0(\mathbf{X}) : \mathbf{\Psi} \in \mathcal{F}_{i+1}\}, \epsilon_{i+1}, |\cdot|_{i+1}),
\end{aligned} \tag{3}$$

*where when $i = 0$, the first term in the product is given by $\mathcal{N}(\{\mathbf{\Psi}(\mathbf{X}) : \mathbf{\Psi} \in \mathcal{F}_1\}, \epsilon_1, |\cdot|_1)$.*

Proposition 1 establishes an upper bound on $\mathcal{N}(\mathcal{H}_L(\mathbf{X}), s_L, |\cdot|_L)$ in terms of the covering numbers

$$\mathcal{N}(\{\mathbf{\Psi} \circ \mathbf{\Psi}_i \circ \mathbf{\Psi}_{i-1} \circ \cdots \circ \mathbf{\Psi}_0(\mathbf{X}) : \mathbf{\Psi} \in \mathcal{F}_{i+1}\}, \epsilon_{i+1}, |\cdot|_{i+1}).$$

It is built upon an iterative argument that generalizes those for MLPs (Anthony et al., 1999; Bartlett et al., 2017) and is adapted to various operator norms (i.e., $|\cdot|_i$) at each layer. To better understand these quantities, we will start by presenting a result when the activation functions can be written

as linear combinations of a set of basis functions. This is a common practice, and the literature has investigated several choices of basis functions, such as B-splines, wavelets, radial basis, Fourier series, finite basis, etc.; see, e.g., Liu et al. (2024c); Seydi (2024b); Li (2024); Xu et al. (2024a); Howard et al. (2024). The proof of the following result relies on Maurey's sparsification lemma stated in the appendix; also see Lemma 1 of Zhang (2002).

**Proposition 2.** *Suppose* $\mathbf{\Psi}(\mathbf{x}) = (\psi_1(\mathbf{x}), \ldots, \psi_m(\mathbf{x}))$ *with* $\psi_i(\mathbf{x}) = \sum_{j=1}^{p} \beta_{ij} g_{ij}(\mathbf{x})$ *for* $1 \leq i \leq m$. *Here* $\{g_{ij} : i = 1, 2, \ldots, m; j = 1, 2, \ldots, p\}$ *is a set of basis functions defined on* $\mathbb{R}^d$. *Let* $\mathbf{B} = (\beta_{ij}) \in \mathbb{R}^{m \times p}$ *and* $\mathbf{G}(\mathbf{X}) = (g_{ij}(\mathbf{x}_k)) \in \mathbb{R}^{m \times p \times n}$. *Define* $\|\mathbf{B}\|_r = (\sum_{i=1}^{m} \sum_{j=1}^{p} |\beta_{ij}|^r)^{1/r}$ *and*

$$\|\mathbf{G}(\mathbf{X})\|_{v,s} = \left\{ \sum_{i=1}^{m} \sum_{j=1}^{p} \left( \sum_{k=1}^{n} |g_{ij}(\mathbf{x}_k)|^v \right)^{s/v} \right\}^{1/s}$$

*for* $0 < v \leq 2$ *and* $r, s > 0$ *with* $1/r + 1/s = 1$. *We assume that* $\|\mathbf{B}\|_r \leq b_{m,p,n}$ *and* $\|\mathbf{G}(\mathbf{X})\|_{v,s} \leq c_{m,p,n}$, *where* $b_{m,p,n}, c_{m,p,n}$ *are both allowed to grow with* $m, p$ *and* $n$. *Then, we have*

$$\log \mathcal{N}\left( \left\{ \mathbf{\Psi}(\mathbf{X}) \in \mathbb{R}^{n \times m} : \mathbf{\Psi} \in \mathcal{F} \right\}, \epsilon, \| \cdot \|_2 \right) \leq \frac{b_{m,p,n}^2 c_{m,p,n}^2}{\epsilon^2} \log(2mp).$$

With the above results, we now provide a bound on the covering number of $\mathcal{H}_L(\mathbf{X})$. We make the following assumptions.

**Assumption 1.** *Assume that* $\|\mathbf{X}\|_2 \leq D$ *for some constant* $D > 0$.

**Assumption 2.** *Suppose* $\mathbf{\Psi}_l$ *belongs to the following function space*

$$\mathcal{F}_l = \left\{ \mathbf{\Psi} = (\psi_1, \ldots, \psi_{d_l}) : \psi_i(\cdot) = \sum_{j=1}^{p_l} \beta_{ij}^{(l)} g_{ij}^{(l)}(\cdot), \|\mathbf{\Psi}(\mathbf{x}) - \mathbf{\Psi}(\mathbf{x}')\|_2 \leq \rho_l \|\mathbf{x} - \mathbf{x}'\|_2, \right.$$

$$\left. \|\mathbf{\Psi}(\mathbf{0})\|_2 \leq C_l, \|\mathbf{B}_l\|_1 = \sum_{i=1}^{d_l} \sum_{j=1}^{p_l} |\beta_{ij}^{(l)}| \leq B_l \right\},$$

*for some constants* $\rho_l, B_l, C_l > 0$. *Further, assume that*

$$|g_{ij}^{(l)}(\mathbf{x}) - g_{ij}^{(l)}(\mathbf{x}')| \leq c_{ij}^{(l)} \|\mathbf{x} - \mathbf{x}'\|_2. \tag{4}$$

We make a set of remarks regarding the above assumptions.

**Remark 1.** Note that the bound on the $l_2$ norm of $\mathbf{X}$ is allowed to grow in our theorem. Moreover, Assumption 1 can be relaxed by requiring the bound to hold with high probability, which can be justified when the distribution of $\mathbf{X}$ has a sufficiently light tail, e.g., sub-Guassian or sub-exponential tails.

**Remark 2.** The form of $\mathbf{\Psi}$ in Assumption 2 is indeed more general than the additive structure in (1), as $g_{ij}^{(l)}$ is a multivariate function. Consider $\mathbf{\Psi} = (\psi_1, \ldots, \psi_{d_l})$, where

$$\psi_i(\mathbf{x}) = \sum_{j=1}^{d_{l-1}} \psi_{i,j}(x_j), \quad \psi_{i,j}(x_j) = \sum_{k=1}^{b_l} \beta_{i,j,k} g_{j,k}(x_j), \tag{5}$$

and $g_{j,k}$ is a set of univariate basis functions. This can be viewed as a special form of the activation function specified in Assumption 2, where the set of $p_l$ basis functions $g_{ij}^{(l)}$ is given by $\{g_{j,k} : 1 \leq j \leq d_{l-1}, 1 \leq k \leq b_l\}$.

**Remark 3.** When $\{g_{ij}^{(l)}\}_j$ is a set of univariate B-spline basis functions with degree $p$ and the knots $\{\xi_{i,j}^{(l)}\}_j$, we have $c_{ij}^{(l)} \leq 2p/\Delta_i^{(l)}$, where $\Delta_i^{(l)} = \max_j(\xi_{i,j+p}^{(l)} - \xi_{i,j}^{(l)})$.

**Remark 4.** Let $\|\mathbf{B}_l\|_0 = |\{\beta_{ij}^{(l)} \neq 0 : 1 \leq i \leq d_l, 1 \leq j \leq p_l\}|$ be the number of non-zero coefficients in $\mathbf{B}_l$. Suppose $\max_{i,j} |\beta_{ij}^{(l)}| \leq B_{\max}$. Then we have $\|\mathbf{B}_l\|_1 \leq B_{\max} \|\mathbf{B}_l\|_0$.

**Remark 5.** The tightest Lipschitz constant for $\mathbf{\Psi}$ is given by

$$\rho^* = \sup_{\mathbf{x} \neq \mathbf{x}'} \frac{\|\mathbf{\Psi}(\mathbf{x}) - \mathbf{\Psi}(\mathbf{x}')\|_2}{\|\mathbf{x} - \mathbf{x}'\|_2}.$$

When $\mathbf{\Psi}(\mathbf{x}) = \mathbf{A}\mathbf{x} + \mathbf{b}$ is linear, $\rho^*$ is simply the spectral norm of the matrix $\mathbf{A}$ (denoted by $\|\mathbf{A}\|_\sigma$). When $\mathbf{\Psi}$ takes the form in (5), we can write $\mathbf{\Psi}(\mathbf{x}) = \mathbf{A}\mathbf{g}(\mathbf{x})$, where $\mathbf{A} \in \mathbb{R}^{d_l \times (d_{l-1}b_l)}$ with the $(i, (j-1)b_l+k)$th element of $\mathbf{A}$ being $\beta_{i,j,k}$, and $\mathbf{g}(\mathbf{x}) \in \mathbb{R}^{d_{l-1}b_l}$ with the $((j-1)b_l+k)$th element of $\mathbf{g}(\mathbf{x})$ being $g_{j,k}(x_j)$. We can deduce that $\rho^* \leq \|\mathbf{A}\|_\sigma \sqrt{\sum_{k=1}^{b_l} a_k^2}$ if $|g_{j,k}(x) - g_{j,k}(x')| \leq a_k |x - x'|$. Moreover, under Condition (4), $\rho^* \leq \|\mathbf{A}\|_\sigma c_l \sqrt{b_l}$, where we define $c_l = \max_{i,j} c_{ij}^{(l)}$.

Let $C = \max_l C_l$. Further, define $\tilde{\alpha} = \sum_{i=1}^L \alpha_i$ with

$$\alpha_i = B_i^{2/3} c_i^{2/3} \left( \prod_{j=i+1}^L \rho_j \right)^{2/3} \left( C \sum_{j=0}^{i-1} \prod_{k=i-j+1}^i \rho_k + D \prod_{k=1}^i \rho_k \right)^{2/3}.$$

The value of $\tilde{\alpha}$ indicates how complex the network is. It is influenced by the $l_1$ norm of the coefficient matrices (i.e., $B_i$) and is also affected by the product of the Lipschitz constants ($\rho_l$) for the activation functions at each layer. Note that for MLPs, the corresponding Lipschitz constants are equal to the spectral norms of the weight matrices. When $C = 0$ (i.e., $C_l = 0$ for all $1 \leq l \leq L$), $\alpha_i = (B_i c_i D \prod_{j=1}^L \rho_j)^{2/3}$ and

$$\tilde{\alpha} = \left( D \prod_{j=1}^L \rho_j \right)^{2/3} \sum_{i=1}^L (B_i c_i)^{2/3}.$$

**Theorem 1.** *Under Assumptions 1-2,*

$$\log \mathcal{N}(\mathcal{H}_L(\mathbf{X}), \epsilon, \|\cdot\|_2) \leq \frac{\tilde{\alpha}^3 \log(2\tilde{d}\tilde{p})}{\epsilon^2},$$

*where $\tilde{d} = \max_i d_i$ and $\tilde{p} = \max_i p_i$.*

It is worth noting that the above bound has no dependence on combinatorial parameters such as the number of nodes and the number of basis functions for each activation function outside of the logarithmic factor. Built upon the above results, we now derive a bound on the generalization error $R(f)$. We impose the following assumption on the loss function.

**Assumption 3.** *Given $v \in \mathcal{Y}$, $\mathcal{L}(\cdot, v)$ is Lipschitz in the sense that $|\mathcal{L}(u, v) - \mathcal{L}(u', v)| \leq B(v)\|u - u'\|_2$ for any $u, u' \in \mathcal{Y}$ and $B(\cdot) : \mathcal{Y} \to \mathbb{R}_{>0}$. Further suppose $\mathcal{L}(f(\cdot), \cdot) \in [0, M]$ for any $f \in \mathcal{M}$.*

**Theorem 2.** *Under Assumptions 1-3, we have with probability greater than $1 - \epsilon$,*

$$R(f) \leq \frac{1}{n} \sum_{i=1}^n \mathcal{L}(f(\mathbf{x}_i), y_i) + \frac{144\sqrt{\zeta}\{\log(nM/(3\sqrt{\zeta})) \vee 1\}}{n}$$

$$+ \sqrt{\frac{4M^2 \log(2/\epsilon)}{n}} + \frac{32M \log(2/\epsilon)}{3n}$$

*for any $f \in \mathcal{M}$, where $\zeta = \tilde{\alpha}^3 \log(2\tilde{d}\tilde{p}) \max_i B^2(y_i)$.*

In Section A.1, we compare this bound with a corresponding bound for MLPs in Bartlett et al. (2017).

Assumption 3 requires the loss to be bounded, which is satisfied by the ramp loss in multiclass classification. However, this excludes many other unbounded loss functions. By using a truncation argument, we can relax the boundedness condition on $\mathcal{L}$ in Assumption 3. This relaxation would allow the result below to cover a general class of nonparametric regression-type loss functions, including squared loss, pinball loss, and Huber loss.

**Assumption 4.** *Given $v \in \mathcal{Y}$, $\mathcal{L}(\cdot, v)$ is Lipschitz in the sense that $|\mathcal{L}(u, v) - \mathcal{L}(u', v)| \leq B(v)\|u - u'\|_2$ for any $u, u' \in \mathcal{Y}$ and $B(\cdot) : \mathcal{Y} \to \mathbb{R}_{>0}$. Further suppose $\sup_{f \in \mathcal{M}} |\mathcal{L}(f(\cdot), \cdot)| \leq G(\cdot, \cdot)$ and $\mathbb{E}[G^s(\mathbf{x}, y)] < C' < \infty$ for $(\mathbf{x}, y) \sim P$ and some $s > 1$, and $\mathbb{E}[B^{s'}(y_i)] < C'' < \infty$ for $s' > 0$.*

**Theorem 3.** *Under Assumptions 1,2 and 4, we have with probability greater than $1 - \epsilon - \tau - \eta$,*

$$R(f) \leq \frac{1}{n} \sum_{i=1}^{n} \mathcal{L}(f(\mathbf{x}_i), y_i) + \frac{144\sqrt{\zeta_0}\{\log(n^{(2s+1)/(2s)}/(3\sqrt{\zeta_0})) \vee 1\}}{n}$$
$$+ \frac{2\sqrt{\log(2/\epsilon)}}{n^{(s-1)/(2s)}} + \frac{32\log(2/\epsilon)}{3n^{(2s-1)/(2s)}} + \frac{2C'}{\eta n^{(s-1)/(2s)}}$$

*for $\epsilon, \tau, \eta > 0$ and any $f \in \mathcal{M}$, where $\zeta_0 = \tilde{\alpha}^3 \log(2\tilde{d}\tilde{p})(nC''/\tau)^{2/s'}$.*

As an implication of the above theorem, consider the empirical risk minimization problem in (2), where $\mathcal{M} = \{f(\cdot) = \boldsymbol{\Psi}_L \circ \boldsymbol{\Psi}_{L-1} \circ \cdots \circ \boldsymbol{\Psi}_1(\cdot) : \boldsymbol{\Psi}_l \in \mathcal{F}_l, l = 1, 2, \ldots, L\}$ with $\mathcal{F}_l$ specified in Assumption 2. Due to the uniformity over the class $\mathcal{M}$, the risk bound in Theorem 3 holds for any solution to Problem (2) resulting from an optimization scheme (assuming that the resulting solution still belongs to $\mathcal{M}$). In Section A.2, we derive an improved high probability bound when $G(\mathbf{x}, y)$ and $B(y)$ both have sub-exponential tails.

Let $f^* = \operatorname{argmin}_{f \in \mathcal{M}} \mathbb{E}[\mathcal{L}(f(\mathbf{x}), y)]$ for $(\mathbf{x}, y) \sim P$ and $R(f^*)$ be the corresponding generalization error. As a byproduct of Theorem 3, we obtain a bound on the excess risk.

**Corollary 1.** *Let $\widehat{f} \in \mathcal{M}$ satisfy that $\sum_{i=1}^{n} \mathcal{L}(\widehat{f}(\mathbf{x}_i), y_i) \leq \sum_{i=1}^{n} \mathcal{L}(f^*(\mathbf{x}_i), y_i)$. Suppose Assumptions 1,2 and 4 hold, where $s \geq 2$ in Assumption 4. We have with probability greater than $1 - \epsilon - \tau - 2\eta$,*

$$R(\widehat{f}) - R(f^*) \leq \frac{144\sqrt{\zeta_0}\{\log(n^{(2s+1)/(2s)}/(3\sqrt{\zeta_0})) \vee 1\}}{n} + (1 + \eta^{-1/2})\sqrt{\frac{2(C')^{2/s}\log(2/\epsilon)}{n}}$$
$$+ \frac{32\log(2/\epsilon)}{3n^{(2s-1)/(2s)}} + \frac{2C'}{\eta n^{(s-1)/(2s)}},$$

*for $\epsilon, \tau, \eta > 0$, where $\zeta_0 = \tilde{\alpha}^3 \log(2\tilde{d}\tilde{p})(nC''/\tau)^{2/s'}$.*

The condition $\sum_{i=1}^{n} \mathcal{L}(\widehat{f}(\mathbf{x}_i), y_i) \leq \sum_{i=1}^{n} \mathcal{L}(f^*(\mathbf{x}_i), y_i)$ only requires $\widehat{f}$ to have an empirical loss smaller than that induced by $f^*$ and it does not necessitate that $\widehat{f}$ be the global minimizer of Problem (2).

### 2.3 RISK ANALYSIS: ACTIVATION FUNCTIONS IN LOW-RANK SOBOLEV SPACE

This subsection studies the case where the activation function $\boldsymbol{\Psi}_l$ belongs to an RKHS with a low-rank structure. Note that when $\boldsymbol{\Psi}_l(\mathbf{x}) = \mathbf{A}\mathbf{x} + \mathbf{b}$ is an affine map, the low-rank assumption translates into a low-rank condition on the matrix $\mathbf{A}$.

To proceed, we let $\mathcal{N}_\mathcal{K}(\Omega)$ be an RKHS on $\Omega \subseteq \mathbb{R}^d$ associated with a reproducing kernel $\mathcal{K}$. Denote by $\|\cdot\|_{\mathcal{N}_\mathcal{K}}$ the corresponding RKHS norm. We focus on the scenario where $\mathcal{K}$ is the isotropic Matérn kernel function, i.e.,

$$\mathcal{K}(\mathbf{x}) = \frac{1}{\Gamma(\nu - d/2)2^{\nu - d/2 - 1}} \|\mathbf{x}\|_2^{\nu - d/2} \mathcal{K}_{\nu - d/2}(\|\mathbf{x}\|_2),$$

after a proper reparametrization, where $\mathcal{K}_{\nu - d/2}$ is the modified Bessel function of the second kind, and $\|\cdot\|_2$ denotes the Euclidean norm. The parameter $\nu > d/2$ controls the smoothness of functions in $\mathcal{N}_\mathcal{K}(\Omega)$. Corollary 10.13 in Wendland (2004), and the extension theorem in (DeVore & Sharpley, 1993) imply that the corresponding RKHS coincides with a Sobolev space with smoothness $\nu$. To model vector-valued functions, we consider the Cartesian product of $\mathcal{N}_\mathcal{K}(\Omega)$, denoted by $\mathcal{N}_\mathcal{K}^{\otimes m}(\Omega)$, given by

$$\mathcal{N}_\mathcal{K}^{\otimes m}(\Omega) = \left\{ \boldsymbol{\Psi} = (\psi_1, \ldots, \psi_m) : \psi_i \in \mathcal{N}_\mathcal{K}(\Omega), 1 \leq i \leq m \right\}.$$

Let $\operatorname{span}(\psi_1, \ldots, \psi_m)$ be the linear space spanned by $\{\psi_i\}_{i=1}^m$ and $\dim(\operatorname{span}(\psi_1, \ldots, \psi_m))$ be its dimension. For $\boldsymbol{\Psi} = (\psi_1, \ldots, \psi_m) \in \mathcal{N}_\mathcal{K}^{\otimes m}$, write $\|\boldsymbol{\Psi}\|_{\mathcal{N}_\mathcal{K}^{\otimes m}}^2 = \sum_{i=1}^m \|\psi_i\|_{\mathcal{N}_\mathcal{K}}^2$. Define

$$\mathcal{A}_r = \left\{ \boldsymbol{\Psi} = (\psi_1, \ldots, \psi_m) \in \mathcal{N}_\mathcal{K}^{\otimes m} : \dim(\operatorname{span}(\psi_1, \ldots, \psi_m)) \leq r \right\}$$

and $\mathcal{A}_r(R) = \{\boldsymbol{\Psi} \in \mathcal{A}_r : \|\boldsymbol{\Psi}\|_{\mathcal{N}_{\mathcal{K}}^{\otimes m}} \leq R\}$. Recall that $\boldsymbol{\Psi}(\mathbf{X}) = [\boldsymbol{\Psi}(\mathbf{x}_1), \dots, \boldsymbol{\Psi}(\mathbf{x}_n)]^\top$ and $\|\boldsymbol{\Psi}(\mathbf{X})\|_2^2 = \sum_{i=1}^m \sum_{j=1}^n \psi_i^2(\mathbf{x}_j)$.

We state the following bound on the metric entropy of $\{\boldsymbol{\Psi}(\mathbf{X}) \in \mathbb{R}^{n \times m} : \boldsymbol{\Psi} \in \mathcal{A}_r(R)\}$ whose proof relies on the equivalence between the RKHS and the Sobolev space.

**Proposition 3** (Proposition A.3 of Wang & Zhou (2020)). Under the above setups, we have

$$\log \mathcal{N}\left(\left\{\boldsymbol{\Psi}(\mathbf{X}) \in \mathbb{R}^{n \times m} : \boldsymbol{\Psi} \in \mathcal{A}_r(R)\right\}, \epsilon, \|\cdot\|_2\right)$$

$$\leq mr \log\left(1 + \frac{\widetilde{C}R\sqrt{rn}}{\epsilon}\right) + r\left(\frac{\widetilde{C}R\sqrt{rn}}{\epsilon}\right)^{d/\nu},$$

where $\widetilde{C}$ is a positive constant.

**Assumption 5.** Suppose $\boldsymbol{\Psi}_l$ belongs to the following function space

$$\mathcal{F}_l = \left\{\boldsymbol{\Psi} = (\psi_1, \dots, \psi_{d_l}) \in A_{r_l}(R_l) : \|\boldsymbol{\Psi}(\mathbf{x}) - \boldsymbol{\Psi}(\mathbf{x}')\|_2 \leq \rho_l \|\mathbf{x} - \mathbf{x}'\|_2\right\},$$

where $\rho_l, r_l$ and $R_l$ are some positive constants.

**Proposition 4.** Set $b_i = \widetilde{C}R_i\sqrt{r_i n}$ and $\tilde{b} = \sum_{i=1}^L b_i$. Under Assumption 5, we have

$$\log \mathcal{N}(\mathcal{H}_L(\mathbf{X}), \epsilon, \|\cdot\|_2) \leq \sum_{i=1}^L d_i r_i \left(\frac{\tilde{b} \prod_{j=i+1}^L \rho_j}{\epsilon}\right)^{(d_{i-1}/\nu)\vee 1}.$$

**Theorem 4.** *Define*

$$\xi = \sum_{i=1}^L d_i r_i \left(\max_i B(y_i) \tilde{b} \prod_{j=i+1}^L \rho_j\right)^{(d_{i-1}/\nu)\vee 1},$$

*where $\tilde{b} = \sum_{i=1}^L b_i$ with $b_i = \widetilde{C}R_i\sqrt{r_i n}$. Suppose $\tilde{d} := \max_i d_i > \nu$. Under Assumptions 3 and 5, we have with probability greater than $1 - \epsilon$,*

$$R(f) \leq \frac{1}{n}\sum_{i=1}^n \mathcal{L}(f(\mathbf{x}_i), y_i) + \frac{6\widetilde{C}'(\xi)^{\nu/\tilde{d}}}{n^{(\nu/\tilde{d}+1)/2}(\tilde{d}/\nu - 1)^{\nu/\tilde{d}}}$$

$$+ \sqrt{\frac{4M^2 \log(2/\epsilon)}{n}} + \frac{32M \log(2/\epsilon)}{3n}$$

*for any $f \in \mathcal{M}$ and some constant $\widetilde{C}' > 0$.*

Theorem 4 generalizes the results for low-rank kernel ridge regression to a multi-layer network structure induced by KANs with activation functions belonging to an RKHS. The generalization bound here scales polynomially with the underlying ranks as well as the Lipschitz constants of the activation functions in each layer. Also, it has no explicit dependence on combinatorial parameters.

**Remark 6.** Given a set of (fixed) activation functions $\boldsymbol{\Phi}_l \in \mathcal{N}_{\mathcal{K}}^{\otimes d_l}$ for $1 \leq l \leq L$, we define the function class

$$\widetilde{\mathcal{F}}_l = \left\{\widetilde{\boldsymbol{\Psi}} = \boldsymbol{\Phi}_l + \boldsymbol{\Psi} : \boldsymbol{\Psi} \in A_{r_l}(R_l), \|\widetilde{\boldsymbol{\Psi}}(\mathbf{x}) - \widetilde{\boldsymbol{\Psi}}(\mathbf{x}')\|_2 \leq \rho_l\|\mathbf{x} - \mathbf{x}'\|_2\right\}.$$

Then, the conclusion in Theorem 4 remains true when the activation function in layer $l$ belongs to $\widetilde{\mathcal{F}}_l$. Suppose $\boldsymbol{\Phi}_l$ is an activation obtained from pre-training, and we perform a fine-tuning to find the activation functions for some new downstream tasks. Then, we require the update after the fine-tuning process to lie on a low-rank space $A_{r_l}(R_l)$. This kind of strategy has been shown to be effective for fine-tuning large language models; see, e.g., (Hu et al., 2021).

To conclude this section, we present the following results that are parallel to Theorem 3 and Corollary 1. Let

$$\xi_0 = \sum_{i=1}^L d_i r_i \left\{\left(\frac{nC''}{\tau}\right)^{2/s'} \tilde{b} \prod_{j=i+1}^L \rho_j\right\}^{(d_{i-1}/\nu)\vee 1}, \tag{6}$$

where $\tilde{b} = \sum_{i=1}^L b_i$ with $b_i = \widetilde{C}R_i\sqrt{r_i n}$.

**Theorem 5.** *Under Assumptions 4 and 5, we have with probability greater than $1 - \epsilon - \tau - \eta$,*

$$R(f) \leq \frac{1}{n} \sum_{i=1}^{n} \mathcal{L}(f(\mathbf{x}_i), y_i) + \frac{6\widetilde{C}'(\xi_0)^{\nu/\tilde{d}}}{n^{(\nu/\tilde{d}+1)/2}(\tilde{d}/\nu - 1)^{\nu/\tilde{d}}}$$
$$+ \frac{2\sqrt{\log(2/\epsilon)}}{n^{(s-1)/(2s)}} + \frac{32\log(2/\epsilon)}{3n^{(2s-1)/(2s)}} + \frac{2C'}{\eta n^{(s-1)/(2s)}}$$

*for any $f \in \mathcal{M}$ and $\epsilon, \tau, \eta, \widetilde{C}' > 0$.*

**Corollary 2.** *Let $\widehat{f} \in \mathcal{M}$ satisfy that $\sum_{i=1}^{n} \mathcal{L}(\widehat{f}(\mathbf{x}_i), y_i) \leq \sum_{i=1}^{n} \mathcal{L}(f^*(\mathbf{x}_i), y_i)$. Suppose Assumptions 1,2 and 4 hold, where $s \geq 2$ in Assumption 4. We have with probability greater than $1 - \epsilon - \tau - 2\eta$,*

$$R(\widehat{f}) - R(f^*) \leq \frac{6\widetilde{C}'(\xi_0)^{\nu/\tilde{d}}}{n^{(\nu/\tilde{d}+1)/2}(\tilde{d}/\nu - 1)^{\nu/\tilde{d}}} + (1 + \eta^{-1/2})\sqrt{\frac{2(C')^{2/s}\log(2/\epsilon)}{n}}$$
$$+ \frac{32\log(2/\epsilon)}{3n^{(2s-1)/(2s)}} + \frac{2C'}{\eta n^{(s-1)/(2s)}},$$

*for $\epsilon, \tau, \eta > 0$, where $\xi_0$ is defined in (6).*

**Remark 7.** Combining the arguments from Theorems 3 and 5, we can derive a generalization bound for KANs in the more general case. This applies when the activation functions of certain layers belong to the class in Assumption 2, while the activation functions of other layers have a low-rank structure, e.g., as specified in Remark 6.

## 3 NUMERICAL STUDIES

We used both the simulated and real datasets to demonstrate the relationship between the excess loss (defined as the difference between the test loss and training loss) and the complexity of KANs. We constructed the KANs such that the output of each layer at $\mathbf{0}$ is $\mathbf{0}$, i.e., $\mathbf{\Psi}_i(\mathbf{0}) = \mathbf{0}$. Therefore, we used $(\prod_{j=1}^{L} \rho_j)^{2/3} \sum_{i=1}^{L}(B_i c_i)^{2/3}$, which was proportional to $\tilde{\alpha}$, as the measure of complexity, where the Lipschitz constants $\rho_j$s were estimated by their upper bounds provided in Remark 5.

For the simulation, we consider two examples in Liu et al. (2024c). Specifically, we generated $x_i$ from Unif$(-1, 1)$ independently, and let

$$f_1(x_1, x_2, x_3, x_4) = \exp\left(\frac{1}{2}\left\{\sin(\pi(x_1^2 + x_2^2)) + \sin(\pi(x_3^2 + x_4^2))\right\}\right),$$

$$f_2(x_1, \ldots, x_{100}) = \exp\left\{\frac{1}{100}\sum_{i=1}^{100}\sin^2\left(\frac{\pi x_i}{2}\right)\right\},$$

for the low-dimensional and high-dimensional cases, respectively. We considered the following four setups:

(i) $y = f_1(x_1, x_2, x_3, x_4) \times \exp(\epsilon)$, $\quad \epsilon \sim N(-\log(1.04)/2, \log(1.04))$;

(ii) $y = f_2(x_1, \ldots, x_{100}) \times \exp(\epsilon)$, $\quad \epsilon \sim N(-\log(1.04)/2, \log(1.04))$;

(iii) $P(y = 1) = \dfrac{1}{1 + f_1(x_1, x_2, x_3, x_4)}$, $\quad P(y = 0) = 1 - P(y = 1)$;

(iv) $P(y = 1) = \dfrac{1}{1 + f_2(x_1, \ldots, x_{100})}$, $\quad P(y = 0) = 1 - P(y = 1)$,

where we have chosen the distribution of $\epsilon$ such that the mean and standard deviation of $\exp(\epsilon)$ are equal to 1 and 0.2, respectively. We set the sample sizes of both the training set and test set to be 10,000 for all four datasets. The shape of KAN used for (i) and (iii) was $[4, 50, 100, 50, 1]$, and was $[100, 50, 100, 50, 1]$ for (ii) and (iv). We refer the readers to Section 2.2 of Liu et al. (2024c) for the definition of the shape of a KAN that is represented by an integer array.

We also investigated the MNIST and CIFAR-10 datasets. We used the features extracted from a pre-trained AlexNet model as input for the KANs. The extracted features were 1000-dimensional for both datasets. We employed the KAN with the shape $[1000, 50, 100, 50, 10]$.

We run 1000 epochs for each dataset. The results are shown in Figure 2, where we normalize the values of the complexity measures so that the maximum value of the complexity measure is equal to the last value of the excess loss (see Section A.4 for more details). The plots in Figure 2 illustrate that our proposed measure of complexity for the KAN networks, $\tilde{\alpha}$, tightly correlates with the excess loss in all the cases. It is surprising to observe that the complexity measure curve closely follows the shape of the excess loss. These results highlight the practical relevance of the generalization bounds derived in Section 2. We refer the readers to the appendix for additional numerical results and discussions.

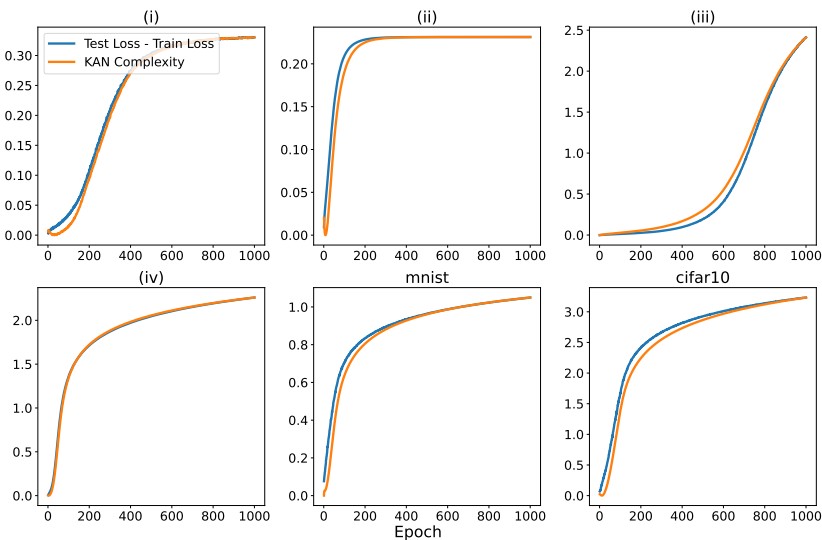

Figure 2: The excess loss and (normalized) complexity of KANs trained with SGD on the four simulated datasets (i–iv) and the MNIST and CIFAR10 datasets. The loss for (iii), (iv), MNIST, and CIFAR10 is the cross-entropy loss, and that for (i) and (ii) is the mean squared error.

## 4    DISCUSSIONS

We derive generalization bounds for KANs when the activation functions are either represented by linear combinations of basis functions or lying in a low-rank RKHS. These bounds are empirically investigated for the KAN networks trained with SGD on simulated and real data sets. The numerical results indicate a strong correlation between the excess loss and the complexity measure, demonstrating the practical relevance of the generalization bounds.

To conclude, we mention a few future research directions. (i) Additional empirical studies and theoretical investigation are needed to understand the superiority of KANs over MLPs; (ii) It would be interesting to derive a lower bound and see how well it matches the upper bound to understand the tightness of the derived bounds. Also, additional work is needed to determine which norms are well-adapted to KANs as used in practice; (iii) A more careful analysis is needed to show that SGD applied to KAN results in a well-behaved predictor and leads to a refined generalization bound; (iv) Theorem 3 may be improved with a relaxed restriction on the Lipschitz constants of the basis functions $\{g_{ij}^{(l)}\}$; (v) It was observed that explicit regularization contributes little to the generalization performance of neural networks (Zhang et al., 2021). It is of interest to see if a similar conclusion holds for KANs and whether other types of regularization help to improve the generalization performance. Our theoretical and empirical results suggest a close connection between the derived complexity measure and the generalization error. It is of interest to develop (implicit) regularization techniques to control this complexity measure to achieve improved generalization performance.

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

# A APPENDIX

## A.1 COMPARISON WITH MLPS

We compare the derived bounds with those for MLPs, specifically focusing on the bounds discussed in Bartlett et al. (2017), which address the classification problem with ramp loss. The scenario described in that work closely aligns with the setting of Theorem 2, particularly with $C = 0$ and $\max_i B(y_i)$ being finite. Ignoring smaller order terms, the bound for $R(f) - n^{1/2} \sum_{i=1}^n \mathcal{L}(f(\mathbf{x}_i), y_i)$ in Theorem 2 can be expressed as follows

$$O\left(\frac{\|\mathbf{X}\| R_{\text{KAN}}}{n} \log(n) \log(2\tilde{d}\tilde{p}) + \sqrt{\frac{\log(1/\epsilon)}{n}}\right),$$

where

$$R_{\text{KAN}} = \left(\prod_{j=1}^L \rho_j\right)\left(\sum_{i=1}^L (B_i c_i)^{2/3}\right)^{3/2}$$

measures the KAN complexity. The corresponding bound for MLPs, derived in Theorem 1.1 of Bartlett et al. (2017), is given by

$$O\left(\frac{\|\mathbf{X}\| R_{\mathcal{A}}}{\gamma n} \log(n) \log(W) + \sqrt{\frac{\log(1/\epsilon)}{n}}\right),$$

where $R_{\mathcal{A}}$ measures model complexity (similar to $R_{\text{KAN}}$ as defined above), $\gamma$ represents the margin in the ramp loss, and $W$ denotes the network width (analogous to $\tilde{d}$ in our case).

When $R$ and $R_{\mathcal{A}}$ are of the same order (denoted by $R \asymp R_{\mathcal{A}}$), both bounds are also of the same order, up to some logarithmic factors. This is expected since $R \asymp R_{\mathcal{A}}$ indicates that both classes exhibit similar levels of complexity. Conversely, if $R = o(R_{\mathcal{A}})$ and $R_{\mathcal{A}}/\sqrt{n} \to +\infty$, then the bound for KANs is of a smaller order than that for MLPs. This scenario arises when the true underlying regression function can be more accurately approximated using KANs. For instance, when the true regression function can be exactly represented by a KAN network, the complexity required for approximating such a function using MLPs is expected to be higher. As discussed in Liu et al. (2024c), KANs may require fewer parameters than MLPs to approximate the same underlying function, as KANs utilize the inherently low-dimensional representation of the true function, unlike MLPs. We refer the readers to Liu et al. (2024c) for examples where KANs can outperform MLPs with the same model complexity, e.g., Figures 3.1-3.3 therein.

## A.2 IMPROVED BOUNDS UNDER SUB-EXPONENTIAL TAILS

We derive an improved high probability bound for the generalization errors under the condition that $G(\mathbf{x}, y)$ and $B(y)$ both have sub-exponential tails. We impose the following assumption.

**Assumption 6.** *Given $v \in \mathcal{Y}$, $\mathcal{L}(\cdot, v)$ is Lipschitz in the sense that $|\mathcal{L}(u, v) - \mathcal{L}(u', v)| \leq B(v)\|u - u'\|_2$ for any $u, u' \in \mathcal{Y}$ and $B(\cdot) : \mathcal{Y} \to \mathbb{R}_{>0}$. Further suppose $\sup_{f \in \mathcal{M}} |\mathcal{L}(f(\cdot), \cdot)| \leq G(\cdot, \cdot)$. Both $G(\mathbf{x}, y)$ and $B(y)$ have sub-exponential tails, i.e., $\mathbb{E}[\exp(\lambda_G G(\mathbf{x}, y))] \leq \exp(\lambda_G^2 \sigma_G^2/2)$ and $\mathbb{E}[\exp(\lambda_B B(y))] \leq \exp(\lambda_B^2 \sigma_B^2/2)$ for $\lambda_G, \lambda_B$ in a neighborhood of 0.*

**Theorem 6.** *Under Assumptions 1,2 and 6, we have with probability greater than $1 - \epsilon - \tau - \eta$,*

$$R(f) \leq \frac{1}{n} \sum_{i=1}^n \mathcal{L}(f(\mathbf{x}_i), y_i) + \frac{144\sqrt{\zeta_0}\{\log(C'n\log(n)/\sqrt{\zeta_0}) \vee 1\}}{n}$$

$$+ \frac{C'\log(n)\sqrt{\log(2/\epsilon)}}{\sqrt{n}} + \frac{C'\log(n)\log(2/\epsilon)}{3n} + C'\sqrt{\frac{\log(1/\eta)}{n}},$$

*for $\epsilon, \tau, \eta > 0$ and any $f \in \mathcal{M}$, where $\zeta_0 = \tilde{\alpha}^3 \log(2\tilde{d}\tilde{p})(C'\log(nC'/\tau))^2$ and $C'$ is a large enough constant.*

Note that the dependency $1/\eta$ in the original bound has been improved to $\sqrt{\log(1/\eta)}$ by using Bernstein's inequality. Additionally, the term $n^{(1-s)/(2s)}$ has been refined to $n^{-1/2}$ due to the exponential tails. In a similar fashion, we have the following result, which strengthens Corollary 1.

**Corollary 3.** *Let $\widehat{f} \in \mathcal{M}$ satisfy that $\sum_{i=1}^{n} \mathcal{L}(\widehat{f}(\mathbf{x}_i), y_i) \leq \sum_{i=1}^{n} \mathcal{L}(f^*(\mathbf{x}_i), y_i)$. Suppose Assumptions 1,2 and 6 hold. We have with probability greater than $1 - \epsilon - \tau - 2\eta$,*

$$R(\widehat{f}) - R(f^*) \leq \frac{144\sqrt{\zeta_0}\{\log(C'n\log(n)/\sqrt{\zeta_0}) \vee 1\}}{n} + \frac{C'\log(n)\sqrt{\log(2/\epsilon)}}{\sqrt{n}}$$

$$+ \frac{C'\log(n)\log(2/\epsilon)}{3n} + C'\sqrt{\frac{\log(1/\eta)}{n}},$$

*for $\epsilon, \tau, \eta > 0$, where $\zeta_0 = \tilde{\alpha}^3 \log(2\tilde{d}\tilde{p})(C'\log(nC'/\tau))^2$ and $C'$ is a large enough constant.*

We remark that the bounds in Theorem 5 and Corollary 2 can also be enhanced under the condition of sub-exponential tails for $G(\mathbf{x}, y)$ and $B(y)$.

### A.3 A LOWER BOUND ON THE EMPIRICAL RADEMACHER COMPLEXITY

We derive a lower bound on the empirical Rademacher complexity defined in (7) for the network class specified in Assumption 2. From the second layer, suppose the basis functions include the projection onto the first coordinate, i.e., there exists a linear combination of the basis such that $\sum_j \beta_{ij}^{(l)} g_{ij}^{(l)}(\mathbf{x}) = x_1$ for $\mathbf{x} = (x_1, \ldots, x_{d_{l-1}})$. For the first layer, we set $\psi_1^{(1)}(\mathbf{x}) = \sum_{j=1}^{p_1} \beta_{1j}^{(1)} g_{1j}^{(1)}(\mathbf{x})$ and $\psi_i^{(1)}(\mathbf{x}) = 0$ for all $i \geq 2$. Let $\psi_1^{(1)}(\mathbf{x}) = x_1$ and Let $\psi_i^{(l)}(\mathbf{x}) = 0$ for $i \geq 2$ and $1 \leq l \leq L - 1$. The activation function for the output layer is a real-valued function that maps the input to its first coordinate. Note that the condition $|\psi_1^{(1)}(\mathbf{x}) - \psi_1^{(1)}(\mathbf{x}')| \leq \rho\|\mathbf{x} - \mathbf{x}'\|_2$ is implied by $\sum_{j=1}^{p_1} |\beta_{1j}^{(1)}| \leq \rho/\max_j c_{1j}^{(1)}$ as $|g_{1j}^{(1)}(\mathbf{x}) - g_{1j}^{(1)}(\mathbf{x}')| \leq c_{1j}^{(1)}\|\mathbf{x} - \mathbf{x}'\|_2$. Therefore, $\boldsymbol{\Psi}_L \circ \cdots \circ \boldsymbol{\Psi}_1(\mathbf{x}) = \sum_{j=1}^{p_1} \beta_{1j}^{(1)} g_{1j}^{(1)}(\mathbf{x})$ and the Rademacher complexity of the network class is lower bounded by that of the class

$$\mathfrak{G}_1 = \left\{ \psi(\mathbf{x}) = \sum_{j=1}^{p_1} \beta_{1j}^{(1)} g_{1j}^{(1)}(\mathbf{x}) : \left( \sum_{j=1}^{p_1} |\beta_{1j}^{(1)}|^2 \right)^{1/2} \leq \widetilde{B}_1 \right\}$$

for $\widetilde{B}_1 = \min\{B_1, \rho/\max_j c_{1j}^{(1)}\}/\sqrt{p_1}$. Then we have

$$\mathcal{R}(\mathfrak{G}_1(\mathbf{X})) = \frac{1}{n}\mathbb{E}\left[ \sup_{\psi \in \mathfrak{G}_1} \sum_{i=1}^{n} e_i \psi(\mathbf{x}_i) \Big| \mathbf{X} \right]$$

$$= \frac{1}{n}\mathbb{E}\left[ \sup_{\boldsymbol{\beta}:\|\boldsymbol{\beta}\|_2 \leq \widetilde{B}_1} \sum_{i=1}^{n} e_i \sum_{j=1}^{p_1} \beta_j g_{1j}^{(1)}(\mathbf{x}_i) \Big| \mathbf{X} \right]$$

$$= \frac{\widetilde{B}_1}{n}\mathbb{E}\left[ \left\| \sum_{i=1}^{n} e_i g_{1j}^{(1)}(\mathbf{x}_i) \right\|_2 \Big| \mathbf{X} \right] \geq \frac{C'\widetilde{B}_1\{\sum_{i=1}^{n} (g_{1j}^{(1)}(\mathbf{x}_i))^2\}^{1/2}}{n},$$

for some constant $C' > 0$, where we have used the Khintchine inequality to get the last inequality.

### A.4 ADDITIONAL NUMERICAL RESULTS

In our experiments, the activation functions in the implemented KAN are linear combinations of SiLU function and basis splines, as proposed by Liu et al. (2024c). Specifically, the $\psi_{i,k,j}(x_j)$ for $i = 1, \ldots, L, k = 1, \ldots, d_i, j = 1, \ldots, d_{i-1}$ in (1) is

$$\psi_{i,k,j}(x) = w_{i,k,j}^{(b)} b(x) + w_{i,k,j}^{(s)} \text{spline}(x),$$

where

$$b(x) = \text{silu}(x) = x/(1 + e^{-x}), \quad \text{spline}(x) = \sum_i c_i B_i(x),$$

and $B_i(x)$ are spline basis. See equations (2.10)-(2.12) of Liu et al. (2024c).

In Figure 2, we display the curves of the KAN complexity as measured by $(\prod_{j=1}^{L} \rho_j)^{2/3} \sum_{i=1}^{L} (B_i c_i)^{2/3}$ and the excess loss on the same plot. Since the values of these two terms are on different scales, we normalize the complexity so that the maximum value of the complexity measure is equal to the last value of the excess loss. Specifically, let $u = (u_1, \ldots, u_N)$ be the values of the differences between the test losses and training losses, where $N$ is the number of training epochs, and $v = (v_1, \ldots, v_N)$ be the model complexity corresponding to different training epoch. Let $v_{\max} = \max\{v_1, \ldots, v_N\}$ and $v_{\min} = \min\{v_1, \ldots, v_N\}$. The normalized complexity is $v' = (v'_1, \ldots, v'_N)$ with

$$v'_i = \frac{(v_i - v_{\min})u_N}{v_{\max} - v_{\min}}.$$

We present the curves of the KAN complexity, test loss, and training loss in Figure 3. The aim of

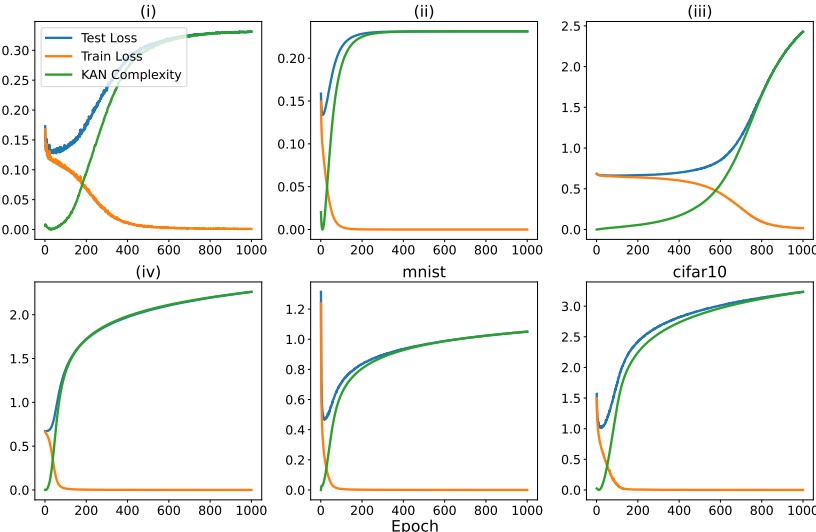

Figure 3: The training loss, test loss and complexity (normalized with respect to test loss) of KANs trained with SGD on the four simulated datasets (i–iv) and the MNIST and CIFAR10 datasets. The loss for (iii), (iv), MNIST, and CIFAR10 is the cross-entropy loss, and that for (i) and (ii) is the mean squared error.

our simulation study is to demonstrate the practical relevance of the generalization bound derived in Section 2. Specifically, we hope to understand the relationship between model complexity, excess risk, and test loss. Our empirical results indicate a strong correlation between excess loss/test loss and model complexity. This finding motivates the need to develop regularization techniques, whether explicit or implicit (such as early stopping or penalization), to control this complexity measure, which can potentially enhance generalization performance.

We use the Setups (i) and (iii) for a simple demonstration, where the underlying regression and classification models are both based on the synthetic function $f_1$ defined in Section 3. We apply the dropout technique with a rate of 0.1 to the activation functions in KAN networks (referred to as regularized KAN) with the same shapes as we have used in Section 3 (i.e., [4, 50, 100, 50, 1] for both setups). The results are shown in the top two panels of Figure 4, where we also plot the results for non-regularized KANs, i.e., those results in Figure 2, for comparison. The values of excess loss are depicted in their original scales, while the values of the KAN complexity are normalized with respect to their excess losses. We observe that with the dropout technique, the excess losses decrease significantly from 0.33 to 0.03 for Setup (i) and from 2.41 to 0.06 for Setup (iii). For both the regularized KAN and non-regularized KAN, we observe that the model complexity tightly correlates with the excess loss. As the two complexity curves are plotted with different normalization scales, they cannot be compared directly. Therefore, we plot the ratios of the complexities of the regularized KAN to the non-regularized KAN in the bottom two panels of Figure 4. We can see that

the regularized KAN, which induces a lower excess loss than the non-regularized KAN, also has a lower complexity.

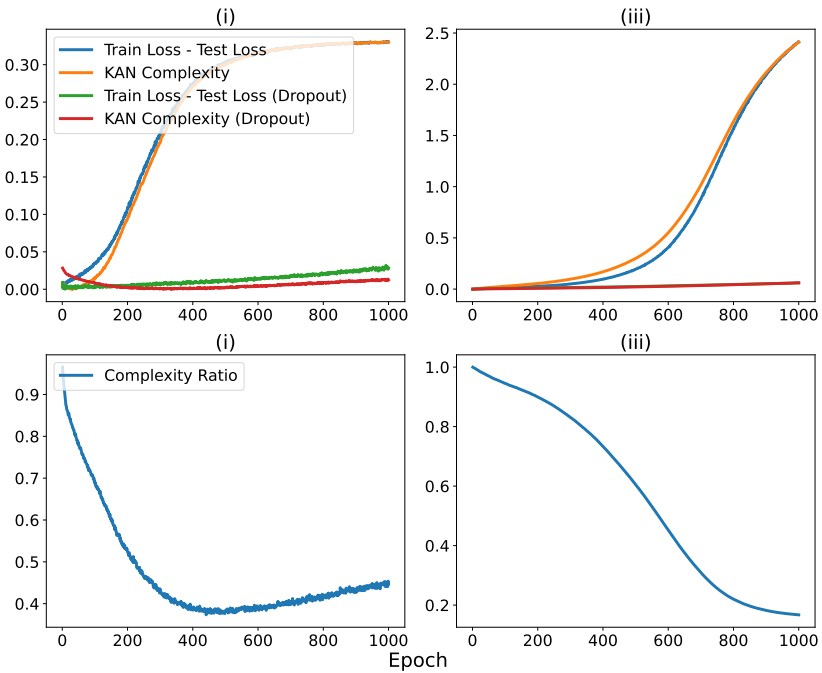

Figure 4: (Top) The excess loss and (normalized) complexity of KANs trained with SGD on the simulated datasets (i) and (iii). The loss for (i) is the mean squared error, and that for (iii) is the cross-entropy loss. The green and red curves in the top-right panel overlap. (Bottom) The ratio of the complexity of the regularized KAN to the non-regularized KAN.

## A.5 Auxiliary Lemmas

The following lemma is useful in the proof of Proposition 2; see Lemma 1 of Zhang (2002) and Lemma A.6 of Bartlett et al. (2017).

**Lemma 1** (Maurey's sparsification lemma). *For a Hilbert space $\mathcal{H}$ with the norm $\|\cdot\|$, let $U \in \mathcal{H}$ be given with the representation $U = \sum_{l=1}^{N} a_l V_l$, where $V_l \in \mathcal{H}$ and $a_l \geq 0$. Then, for any positive integer $k$, there exists a choice of nonnegative integers $(k_1, \ldots, k_N)$ such that $\sum_{i=1}^{N} k_i = k$ and*

$$\left\| U - \frac{\|\mathbf{a}\|_1}{k} \sum_{l=1}^{N} k_l V_l \right\|^2 \leq \frac{\|\mathbf{a}\|_1}{k} \sum_{l=1}^{N} a_l \|V_l\|^2 \leq \frac{\|\mathbf{a}\|_1^2}{k} \max_i \|V_i\|^2,$$

*where $\mathbf{a} = (a_1, \ldots, a_N)$ and $\|\mathbf{a}\|_1 = \sum_{l=1}^{N} a_l$.*

The proof of Theorem 2 makes use of the following two lemmas. The first lemma shows that $R(f)$ can be related to the empirical Rademacher complexity of the neural network function class. The second lemma connects the empirical Rademacher complexity with the corresponding covering number through the integral entropy.

**Lemma 2** (Theorem 2.1 in Bartlett et al. (2005)). *Let $\mathcal{F}$ be a class of functions that maps $\mathcal{X}$ into $[-M, M]$ for some $M > 0$. Assume that there is some $r \geq 0$ such that for every $f \in \mathcal{F}$, $\mathrm{var}(f(\mathbf{x})) \leq r$. Then for every $\epsilon > 0$, with probability at least $1 - \epsilon$ over the data $\mathbf{X} = [\mathbf{x}_1, \ldots, \mathbf{x}_n]^\top$, we have*

$$\sup_{f \in \mathcal{F}} \left\{ \mathbb{E}[f(\mathbf{x})] - \frac{1}{n} \sum_{i=1}^{n} f(\mathbf{x}_i) \right\} \leq 6\mathcal{R}(\mathcal{F}(\mathbf{X})) + \sqrt{\frac{2r \log(2/\epsilon)}{n}} + \frac{32M \log(2/\epsilon)}{3n},$$

where $\mathcal{F}(\mathbf{X}) = \{(f(\mathbf{x}_1), \ldots, f(\mathbf{x}_n)) : f \in \mathcal{F}\}$ and $\mathcal{R}(\mathcal{F}(\mathbf{X}))$ is the empirical Rademacher complexity of $\mathcal{F}$, i.e.,

$$\mathcal{R}(\mathcal{F}(\mathbf{X})) = \frac{1}{n}\mathbb{E}\left[\sup_{f \in \mathcal{F}} \sum_{i=1}^{n} e_i f(\mathbf{x}_i) \Big| \mathbf{X}\right], \tag{7}$$

with $e_i$ being a set of independent Rademacher random variables, i.e., $P(e_i = \pm 1) = 1/2$.

**Lemma 3** (Lemma A.5 in Bartlett et al. (2017)). Let $\mathcal{F}$ be a real-valued function class taking values in $[0, M]$ and assume that $\mathbf{0} \in \mathcal{F}$. Then we have

$$\mathcal{R}(\mathcal{F}(\mathbf{X})) \leq \inf_{a > 0} \left(\frac{4a}{\sqrt{n}} + \frac{12}{n} \int_a^{M\sqrt{n}} \sqrt{\log \mathcal{N}(\mathcal{F}(\mathbf{X}), \epsilon, \|\cdot\|_2)} d\epsilon\right).$$

### A.6 PROOFS OF THE MAIN RESULTS

*Proof of Proposition 1.* We divide the proof into three steps.
**Step 1:** Choose an $\epsilon_1$ cover $\mathcal{N}_1$ of $\{\mathbf{\Psi}(\mathbf{X}) : \mathbf{\Psi} \in \mathcal{F}_1\}$. Then we have

$$|\mathcal{N}_1| \leq \mathcal{N}\left(\{\mathbf{\Psi}(\mathbf{X}) : \mathbf{\Psi} \in \mathcal{F}_1\}, \epsilon_1, |\cdot|_1\right) := N_1.$$

**Step 2:** For every $\mathbf{\Phi} \in \mathcal{N}_i$, we construct an $\epsilon_{i+1}$ cover $\mathcal{G}_{i+1}(\mathbf{\Phi})$ of $\{\mathbf{\Psi}(\mathbf{\Phi}) : \mathbf{\Psi} \in \mathcal{F}_{i+1}\}$. Since the covers are proper, we have $\mathbf{\Phi} = \mathbf{\Psi}_i \circ \mathbf{\Psi}_{i-1} \circ \cdots \circ \mathbf{\Psi}_0(\mathbf{X})$ for some $(\mathbf{\Psi}_i, \ldots, \mathbf{\Psi}_1) \in \mathcal{F}_i \times \mathcal{F}_{i-1} \times \cdots \times \mathcal{F}_1$ and $\mathcal{G}_{i+1}(\mathbf{\Phi}) \subset \mathcal{H}_{i+1}(\mathbf{X})$. Thus, we must have

$$|\mathcal{G}_{i+1}(\mathbf{\Phi})| \leq \sup_{\mathbf{\Psi}_1, \ldots, \mathbf{\Psi}_i} \mathcal{N}(\{\mathbf{\Psi} \circ \mathbf{\Psi}_i \circ \mathbf{\Psi}_{i-1} \circ \cdots \circ \mathbf{\Psi}_0(\mathbf{X}) : \mathbf{\Psi} \in \mathcal{F}_{i+1}\}, \epsilon_{i+1}, |\cdot|_{i+1}) := N_{i+1}.$$

Finally, we form the cover

$$\mathcal{N}_{i+1} = \cup_{\mathbf{\Phi} \in \mathcal{N}_i} \mathcal{G}_{i+1}(\mathbf{\Phi}),$$

which satisfies that

$$|\mathcal{N}_{i+1}| \leq |\mathcal{N}_i||N_{i+1}| \leq \prod_{j=1}^{i+1} N_j.$$

**Step 3:** By the definition of $s_i$, we have $s_i\rho_{i+1} + \epsilon_{i+1} = s_{i+1}$. We shall prove equation 3 using induction. By construction, we can find $\tilde{f}_1 \in \mathcal{N}_1$ such that

$$|\tilde{f}_1(\mathbf{X}) - \mathbf{\Psi}_1 \circ \mathbf{\Psi}_0(\mathbf{X})|_1 \leq \epsilon_1 = s_1.$$

Now suppose we can find $\tilde{f}_i \in \mathcal{N}_i$ such that

$$|\tilde{f}_i(\mathbf{X}) - \mathbf{\Psi}_i \circ \cdots \circ \mathbf{\Psi}_0(\mathbf{X})|_i \leq s_i.$$

By the construction of $\mathcal{N}_{i+1}$, given $\tilde{f}_i \in \mathcal{N}_i$ and $\mathbf{\Psi}_{i+1} \in \mathcal{F}_{i+1}$, we can find $\tilde{f}_{i+1} \in \mathcal{N}_{i+1}$ such that

$$|\tilde{f}_{i+1}(\mathbf{X}) - \mathbf{\Psi}_{i+1} \circ \tilde{f}_i(\mathbf{X})|_{i+1} \leq \epsilon_{i+1}.$$

Then, by the triangle inequality, we have

$$\begin{aligned}
&|\tilde{f}_{i+1}(\mathbf{X}) - \mathbf{\Psi}_{i+1} \circ \mathbf{\Psi}_i \circ \cdots \circ \mathbf{\Psi}_0(\mathbf{X})|_{i+1} \\
\leq &|\tilde{f}_{i+1}(\mathbf{X}) - \mathbf{\Psi}_{i+1} \circ \tilde{f}_i(\mathbf{X})|_{i+1} + |\mathbf{\Psi}_{i+1} \circ \tilde{f}_i(\mathbf{X}) - \mathbf{\Psi}_{i+1} \circ \mathbf{\Psi}_i \circ \cdots \circ \mathbf{\Psi}_0(\mathbf{X})|_{i+1} \\
\leq &\epsilon_{i+1} + \rho_{i+1}|\tilde{f}_i(\mathbf{X}) - \mathbf{\Psi}_i \circ \cdots \circ \mathbf{\Psi}_0(\mathbf{X})|_{i+1} \\
\leq &\epsilon_{i+1} + \rho_{i+1}s_i = s_{i+1}.
\end{aligned}$$

The conclusion thus follows. $\square$

*Proof Proposition 2.* Let $E_{ij}(\mathbf{X}) \in \mathbb{R}^{n \times m}$, where its $i$th column equals $(g_{ij}(\mathbf{x}_1)/b_{ij}, \ldots, g_{ij}(\mathbf{x}_n)/b_{ij})^\top$ with $b_{ij} = (\sum_{k=1}^{n} |g_{ij}(\mathbf{x}_k)|^v)^{1/v}$ and all the other elements are equal to zero. Further, define the set

$$\{V_1, \ldots, V_N\} = \{\tau E_{ij}(\mathbf{X}) : \tau \in \{-1, 1\}, 1 \leq i \leq m, 1 \leq j \leq p\},$$

where $N = 2mp$. We can view $V_i$'s as elements in the Hilbert space $\mathbb{R}^{n \times m}$ equipped with the inner product $\langle A, \widetilde{A} \rangle = \text{trace}(A^\top \widetilde{A})$ and norm $\|A\|_2$ for $A, \widetilde{A} \in \mathbb{R}^{n \times m}$. We have

$$\mathbf{\Psi}(\mathbf{X}) = \sum_{i=1}^{m} \sum_{j=1}^{p} \tau_{ij} b_{ij} |\beta_{ij}| E_{ij}(\mathbf{X}) = \sum_{l=1}^{N} \alpha_l V_l,$$

for $\alpha_l \geq 0$ and $\tau_{ij} = \text{sign}(\beta_{ij})$. Note that

$$\max_{1 \leq i \leq N} \|V_i\|_2 = \max_{1 \leq i \leq m, 1 \leq j \leq p} \frac{(\sum_{k=1}^{n} |g_{ij}(\mathbf{x}_k)|^2)^{1/2}}{(\sum_{k=1}^{n} |g_{ij}(\mathbf{x}_k)|^v)^{1/v}} \leq 1,$$

and

$$\|\mathbf{a}\|_1 = \sum_{l=1}^{N} |a_l| = \sum_{i=1}^{m} \sum_{j=1}^{p} b_{ij} |\beta_{ij}| \leq \|\mathbf{G}(\mathbf{X})\|_{v,s} \|\mathbf{B}\|_r \leq b_{m,p,n} c_{m,p,n},$$

where $1/s + 1/r = 1$ and $s, r > 0$. Therefore, $\mathbf{\Psi}(\mathbf{X}) \in \{b_{m,p,n} c_{m,p,n} \sum_{l=1}^{N} a_l V_l : a_l \geq 0, \sum_l a_l = 1\}$ (this is because the convex hull of $\{V_1, \ldots, V_N\}$ always contains the origin). Now we define

$$\mathcal{C}_k = \left\{ \frac{b_{m,p,n} c_{m,p,n}}{k} \sum_{i=1}^{N} k_i V_i : k_i \geq 0, \sum_{i=1}^{N} k_i = k \right\}$$

$$= \left\{ \frac{b_{m,p,n} c_{m,p,n}}{k} \sum_{j=1}^{k} V_{i_j} : (i_1, \ldots, i_k) \in [N]^k \right\}.$$

By construction, $|\mathcal{C}_k| \leq N^k$. Using Lemma 1, there exists nonnegative integers $(k_1, \ldots, k_N)$ with $\sum_{i=1}^{N} k_i = k$ such that

$$\left\| \mathbf{\Psi}(\mathbf{X}) - \frac{b_{m,p,n} c_{m,p,n}}{k} \sum_{i=1}^{N} k_i V_i \right\|_2^2 \leq \frac{b_{m,p,n}^2 c_{m,p,n}^2}{k} \max_{1 \leq i \leq N} \|V_i\|^2 = \frac{b_{m,p,n}^2 c_{m,p,n}^2}{k}.$$

By setting $\epsilon^2 = b_{m,p,n}^2 c_{m,p,n}^2 / k$, $\mathcal{C}_k$ forms an $\epsilon$ cover of $\{\mathbf{\Psi}(\mathbf{X}) \in \mathbb{R}^{n \times m} : \mathbf{\Psi} \in \mathcal{F}\}$ and the result follows. $\qquad \square$

*Proof of Theorem 1.* Given $(\mathbf{\Psi}_1, \ldots, \mathbf{\Psi}_{l-1}) \in \mathcal{F}_1 \times \cdots \times \mathcal{F}_{l-1}$, we define $\mathbf{X}^{(l-1)}(\mathbf{\Psi}_1, \ldots, \mathbf{\Psi}_{l-1}) = [\mathbf{x}_1^{(l-1)}, \ldots, \mathbf{x}_n^{(l-1)}]^\top$, where $\mathbf{x}_k^{(l-1)} = \mathbf{\Psi}_{l-1} \circ \mathbf{\Psi}_{l-2} \circ \cdots \circ \mathbf{\Psi}_0(\mathbf{x}_k)$ for $1 \leq k \leq n$. Then, we have

$$\|\mathbf{G}^{(l)}(\mathbf{X}^{(l-1)}(\mathbf{\Psi}_1, \ldots, \mathbf{\Psi}_{l-1})) - \mathbf{G}^{(l)}(\mathbf{0})\|_{\infty,2} = \max_{i,j} \left( \sum_{k=1}^{n} |g_{ij}^{(l)}(\mathbf{x}_k^{(l-1)}) - g_{ij}^{(l)}(\mathbf{0})|^2 \right)^{1/2}$$

for $\mathbf{G}^{(l)}(\mathbf{X}) = (g_{ij}^{(l)}(\mathbf{x}_k)) \in \mathbb{R}^{d_l \times p_l \times n}$. We note that

$$\|\mathbf{G}^{(l)}(\mathbf{X}^{(l-1)}(\mathbf{\Psi}_1, \ldots, \mathbf{\Psi}_{l-1})) - \mathbf{G}^{(l)}(\mathbf{0})\|_2$$
$$\leq c_l \|\mathbf{\Psi}_l(\mathbf{\Psi}_{l-1} \circ \cdots \circ \mathbf{\Psi}_0(\mathbf{X}))\|_2$$
$$\leq c_l \left( \|\mathbf{\Psi}_l(\mathbf{\Psi}_{l-1} \circ \cdots \circ \mathbf{\Psi}_0(\mathbf{X})) - \mathbf{\Psi}_l(\mathbf{0})\|_2 + \|\mathbf{\Psi}_l(\mathbf{0})\|_2 \right)$$
$$\leq c_l \left( \rho_l \|\mathbf{\Psi}_{l-1} \circ \cdots \circ \mathbf{\Psi}_0(\mathbf{X})\|_2 + \|\mathbf{\Psi}_l(\mathbf{0})\|_2 \right)$$
$$\leq c_l \left( \sum_{j=0}^{l-1} \|\mathbf{\Psi}_{l-j}(\mathbf{0})\|_2 \prod_{i=l-j+1}^{l} \rho_i + \|\mathbf{X}\|_2 \prod_{i=1}^{l} \rho_i \right)$$
$$\leq c_l \left( C \sum_{j=0}^{l-1} \prod_{i=l-j+1}^{l} \rho_i + D \prod_{i=1}^{l} \rho_i \right).$$

By Proposition 1, we have

$$\log \mathcal{N}(\mathcal{H}_L(\mathbf{X}), s_L, \|\cdot\|_2)$$

$$\leq \sum_{i=0}^{L-1} \sup_{\mathbf{\Psi}_1,\ldots,\mathbf{\Psi}_i} \log \mathcal{N}(\{\mathbf{\Psi} \circ \mathbf{\Psi}_i \circ \mathbf{\Psi}_{i-1} \circ \cdots \circ \mathbf{\Psi}_0(\mathbf{x}) : \mathbf{\Psi} \in \mathcal{F}_{i+1}\}, \epsilon_{i+1}, \|\cdot\|_2)$$

$$= \sum_{i=0}^{L-1} \sup_{\mathbf{\Psi}_1,\ldots,\mathbf{\Psi}_i} \log \mathcal{N}(\{\bar{\mathbf{\Psi}} \circ \mathbf{\Psi}_i \circ \mathbf{\Psi}_{i-1} \circ \cdots \circ \mathbf{\Psi}_0(\mathbf{x}) : \mathbf{\Psi} \in \mathcal{F}_{i+1}\}, \epsilon_{i+1}, \|\cdot\|_2)$$

$$\leq \sum_{i=1}^{L} \frac{\|B_i\|_1^2 c_i^2 \left(C \sum_{j=0}^{i-1} \prod_{k=i-j+1}^{i} \rho_k + D \prod_{k=1}^{i} \rho_k\right)^2}{\epsilon_i^2} \log(2d_i p_i)$$

where $\bar{\mathbf{\Psi}}(\cdot) = \mathbf{\Psi}(\cdot) - \mathbf{\Psi}(\mathbf{0})$. For any $\epsilon > 0$, set

$$\epsilon_i = \frac{\alpha_i \epsilon}{\tilde{\alpha} \prod_{j=i+1}^{L} \rho_j}.$$

Then, we have $s_L = \sum_{i=1}^{L} \left(\prod_{j=i+1}^{L} \rho_j\right) \epsilon_i = \epsilon$ and hence

$$\log \mathcal{N}(\mathcal{H}_L(\mathbf{X}), \epsilon, \|\cdot\|_2) \leq \frac{\tilde{\alpha}^3 \log(2\tilde{d}\tilde{p})}{\epsilon^2}.$$

$\square$

*Proof of Theorem 2.* Define the class $\mathcal{M}_{\mathcal{L}}(\mathbf{X}) = \{(\mathcal{L}(f(\mathbf{x}_1), y_1), \ldots, \mathcal{L}(f(\mathbf{x}_n), y_n)) : f \in \mathcal{M}\}$. By Assumption 3, we have

$$\log \mathcal{N}(\mathcal{M}_{\mathcal{L}}(\mathbf{X}), \epsilon, \|\cdot\|_2) \leq \frac{\tilde{\alpha}^3 \log(2\tilde{d}\tilde{p}) \max_i B^2(y_i)}{\epsilon^2} = \frac{\zeta}{\epsilon^2}.$$

Lemma 3 implies that

$$\mathcal{R}(\mathcal{M}_{\mathcal{L}}(\mathbf{X})) \leq \inf_{a>0} \left(\frac{4a}{\sqrt{n}} + \frac{12}{n} \int_a^{\sqrt{n}M} \sqrt{\frac{\zeta}{\epsilon^2}} d\epsilon\right)$$

$$= \inf_{a>0} \left(\frac{4a}{\sqrt{n}} + \frac{12\sqrt{\zeta}}{n} \log(M\sqrt{n}/a)\right)$$

$$\leq \frac{12\sqrt{\zeta}}{n} + \frac{12\sqrt{\zeta} \log(nM/(3\sqrt{\zeta}))}{n}$$

$$\leq \frac{24\sqrt{\zeta}\{\log(nM/(3\sqrt{\zeta})) \vee 1\}}{n}.$$

Using Lemma 2 with $r = 2M^2$, we have with probability greater than $1 - \epsilon$,

$$R(f) \leq \frac{1}{n} \sum_{i=1}^{n} \mathcal{L}(f(\mathbf{x}_i), y_i) + \frac{144\sqrt{\zeta}\{\log(nM/(3\sqrt{\zeta})) \vee 1\}}{n}$$

$$+ \sqrt{\frac{4M^2 \log(2/\epsilon)}{n}} + \frac{32M \log(2/\epsilon)}{3n},$$

for any $f \in \mathcal{M}$.

$\square$

*Proof of Theorem 3.* Define the class $\mathcal{M}_{\mathcal{L},M}(\mathbf{X}) = \{(\mathcal{L}(f(\mathbf{x}_1), y_1) \wedge M, \ldots, \mathcal{L}(f(\mathbf{x}_n), y_n) \wedge M) : f \in \mathcal{M}\}$. Following the arguments in the proof of Theorem 2 with $\mathcal{M}_{\mathcal{L}}(\mathbf{X})$ replaced by $\mathcal{M}_{\mathcal{L},M}(\mathbf{X})$, we can show that with probability greater than $1 - \epsilon$,

$$\mathbb{E}[\mathcal{L}(f(\mathbf{x}), y) \wedge M] \leq \frac{1}{n} \sum_{i=1}^{n} \mathcal{L}(f(\mathbf{x}_i), y_i) \wedge M + \frac{144\sqrt{\zeta}\{\log(nM/(3\sqrt{\zeta})) \vee 1\}}{n}$$

$$+ \sqrt{\frac{4M^2 \log(2/\epsilon)}{n}} + \frac{32M \log(2/\epsilon)}{3n}.$$

We first note that

$$P(\max_i B^2(y_i) > \varepsilon) \le \sum_{i=1}^n P(B(y_i) > \sqrt{\varepsilon}) \le \frac{nC''}{\varepsilon^{s'/2}}.$$

Setting $\varepsilon = (nC''/\tau)^{2/s'}$, we have $\zeta \le \zeta_0$ with probability greater than $1 - \tau$. Next, under Assumption 4, we have

$$|R(f) - \mathbb{E}[\mathcal{L}(f(\mathbf{x}), y) \wedge M]| \le \mathbb{E}[(\mathcal{L}(f(\mathbf{x}), y) - M)\mathbf{1}\{\mathcal{L}(f(\mathbf{x}), y) > M\}]$$

$$\le \mathbb{E}[G(\mathbf{x}, y)\mathbf{1}\{G(\mathbf{x}, y) > M\}] \le \frac{C'}{M^{s-1}}.$$

Moreover,

$$\left| \frac{1}{n}\sum_{i=1}^n \mathcal{L}(f(\mathbf{x}_i), y_i) - \frac{1}{n}\sum_{i=1}^n \mathcal{L}(f(\mathbf{x}_i), y_i) \wedge M \right| \le \frac{1}{n}\sum_{i=1}^n Z_i,$$

where $Z_i = G(\mathbf{x}_i, y_i)\mathbf{1}\{G(\mathbf{x}_i, y_i) > M\}$. As

$$P\left( \frac{1}{n}\sum_{i=1}^n Z_i > \varepsilon \right) \le \frac{C'}{\varepsilon M^{s-1}},$$

the conclusion follows by setting $\varepsilon = C'/(\eta M^{s-1})$ and $M = n^{1/(2s)}$. $\qquad\square$

**Remark 8.** *When $s \ge 2$, we note that $var(\mathcal{L}(f(\mathbf{x}), y)) \le \mathbb{E}[G^2(\mathbf{x}, y)] \le (\mathbb{E}[G^s(\mathbf{x}, y)])^{2/s} \le (C')^{2/s}$. Thus by Lemma 2 with $r = (C')^{2/s}$, the term $\frac{2\sqrt{\log(2/\epsilon)}}{n^{(s-1)/(2s)}}$ in Theorem 3 can be replaced by $\sqrt{\frac{2(C')^{2/s}\log(2/\epsilon)}{n}}$, leading to the risk-bound*

$$R(f) \le \frac{1}{n}\sum_{i=1}^n \mathcal{L}(f(\mathbf{x}_i), y_i) + \frac{144\sqrt{\zeta_0}\{\log(n^{(2s+1)/(2s)}/(3\sqrt{\zeta_0})) \vee 1\}}{n}$$

$$+ \sqrt{\frac{2(C')^{2/s}\log(2/\epsilon)}{n}} + \frac{32\log(2/\epsilon)}{3n^{(2s-1)/(2s)}} + \frac{2C'}{\eta n^{(s-1)/(2s)}}.$$

*Proof of Corollary 1.* By Remark 8 and the definition of $\widehat{f}$, we have with probability greater than $1 - \epsilon - \tau - \eta$,

$$R(\widehat{f}) \le \frac{1}{n}\sum_{i=1}^n \mathcal{L}(f^*(\mathbf{x}_i), y_i) + \frac{144\sqrt{\zeta_0}\{\log(n^{(2s+1)/(2s)}/(3\sqrt{\zeta_0})) \vee 1\}}{n}$$

$$+ \sqrt{\frac{2(C')^{2/s}\log(2/\epsilon)}{n}} + \frac{32\log(2/\epsilon)}{3n^{(2s-1)/(2s)}} + \frac{2C'}{\eta n^{(s-1)/(2s)}}.$$

Note that

$$P\left( \left| \frac{1}{n}\sum_{i=1}^n \mathcal{L}(f^*(\mathbf{x}_i), y_i) - R(f^*) \right| > \varepsilon \right) \le \frac{var(\mathcal{L}(f^*(\mathbf{x}), y))}{n\epsilon^2} \le \frac{\mathbb{E}[G^2(\mathbf{x}, y)]}{n\epsilon^2} \le \frac{(C')^{2/s}}{n\epsilon^2}.$$

The conclusion follows by setting $\varepsilon = \sqrt{(C')^{2/s}/(n\eta)}$. $\qquad\square$

*Proof of Proposition 4.* The results follow from Propositions 1 and 3. In particular, by Proposition 1, we have

$$\log \mathcal{N}(\mathcal{H}_L(\mathbf{X}), s_L, \|\cdot\|_2)$$

$$\le \sum_{i=0}^{L-1} \sup_{\Psi_1,\dots,\Psi_i} \log \mathcal{N}(\{\mathbf{\Psi} \circ \mathbf{\Psi}_i \circ \mathbf{\Psi}_{i-1} \circ \cdots \circ \mathbf{\Psi}_0(\mathbf{X}) : \mathbf{\Psi} \in \mathcal{F}_{i+1}\}, \epsilon_{i+1}, \|\cdot\|_2),$$

where $\mathcal{F}_{i+1}$ is defined in Assumption 5. By Proposition 3,

$$\log \mathcal{N}(\{\boldsymbol{\Psi} \circ \boldsymbol{\Psi}_i \circ \boldsymbol{\Psi}_{i-1} \circ \cdots \circ \boldsymbol{\Psi}_0(\mathbf{X}) : \boldsymbol{\Psi} \in A_{r_{i+1}}(R_{i+1})\}, \epsilon_{i+1}, \|\cdot\|_2)$$

$$\leq d_{i+1} r_{i+1} \log \left(1 + \frac{\widetilde{C} R_{i+1} \sqrt{r_{i+1}n}}{\epsilon_{i+1}}\right) + r_{i+1} \left(\frac{\widetilde{C} R_{i+1} \sqrt{r_{i+1}n}}{\epsilon_{i+1}}\right)^{d_i/\nu}$$

uniformly over all $\boldsymbol{\Psi}_1, \ldots, \boldsymbol{\Psi}_i$. For any $\epsilon > 0$, set

$$\epsilon_i = \frac{b_i \epsilon}{\tilde{b} \prod_{j=i+1}^{L} \rho_j}.$$

Then, we have $s_L = \sum_{i=1}^{L} \left(\prod_{j=i+1}^{L} \rho_j\right) \epsilon_i = \epsilon$ and for small enough $\epsilon$,

$$\log \mathcal{N}(\mathcal{H}_L(\mathbf{X}), \epsilon, \|\cdot\|_2) \leq \sum_{i=1}^{L} \left\{ d_i r_i \log \left(1 + \frac{\tilde{b} \prod_{j=i+1}^{L} \rho_j}{\epsilon}\right) + r_i \left(\frac{\tilde{b} \prod_{j=i+1}^{L} \rho_j}{\epsilon}\right)^{d_{i-1}/\nu} \right\}$$

$$\leq \sum_{i=1}^{L} d_i r_i \left(\frac{\tilde{b} \prod_{j=i+1}^{L} \rho_j}{\epsilon}\right)^{(d_{i-1}/\nu)\vee 1},$$

where we have used the fact that $\log(1+x) \leq x$ for $x \geq 0$. $\qquad \square$

*Proof of Theorem 4.* Recall $\mathcal{M}_{\mathcal{L}}(\mathbf{X}) = \{(\mathcal{L}(f(\mathbf{x}_1), y_1), \ldots, \mathcal{L}(f(\mathbf{x}_n), y_n)) : f \in \mathcal{M}\}$. By Assumption 3, we have

$$\log \mathcal{N}(\mathcal{M}_{\mathcal{L}}(\mathbf{X}), \epsilon, \|\cdot\|_2) \leq \sum_{i=1}^{L} d_i r_i \left(\frac{\max_i B(y_i) \tilde{b} \prod_{j=i+1}^{L} \rho_j}{\epsilon}\right)^{(d_{i-1}/\nu)\vee 1} \leq \frac{\xi}{\epsilon^{\tilde{d}/\nu}}.$$

Lemma 3 implies that

$$\mathcal{R}(\mathcal{M}_{\mathcal{L}}(\mathbf{X})) \leq \inf_{a>0} \left(\frac{4a}{\sqrt{n}} + \frac{12}{n} \int_a^{\sqrt{n}M} \frac{\xi}{\epsilon^{\tilde{d}/\nu}} d\epsilon\right)$$

$$= \inf_{a>0} \left(\frac{4a}{\sqrt{n}} + \frac{12\xi}{n} \frac{a^{1-\tilde{d}/\nu} - (\sqrt{n}M)^{1-\tilde{d}/\nu}}{\tilde{d}/\nu - 1}\right)$$

$$\leq \frac{\widetilde{C}'(\xi)^{\nu/\tilde{d}}}{n^{(\nu/\tilde{d}+1)/2}(\tilde{d}/\nu - 1)^{\nu/\tilde{d}}}$$

for some constant $\widetilde{C}' > 0$. Using Lemma 2 with $r = 2M^2$, we have with probability greater than $1 - \epsilon$,

$$R(f) \leq \frac{1}{n} \sum_{i=1}^{n} \mathcal{L}(f(\mathbf{x}_i), y_i) + \frac{6\widetilde{C}'(\xi)^{\nu/\tilde{d}}}{n^{(\nu/\tilde{d}+1)/2}(\tilde{d}/\nu - 1)^{\nu/\tilde{d}}}$$

$$+ \sqrt{\frac{4M^2 \log(2/\epsilon)}{n}} + \frac{32M \log(2/\epsilon)}{3n}$$

for any $f \in \mathcal{M}$. $\qquad \square$

*Proof of Theorem 5.* Define the class $\mathcal{M}_{\mathcal{L},M}(\mathbf{X}) = \{(\mathcal{L}(f(\mathbf{x}_1), y_1) \wedge M, \ldots, \mathcal{L}(f(\mathbf{x}_n), y_n) \wedge M) : f \in \mathcal{M}\}$. Following the arguments in the proof of Theorem 4 with $\mathcal{M}_{\mathcal{L}}(\mathbf{X})$ replaced by $\mathcal{M}_{\mathcal{L},M}(\mathbf{X})$, we have with probability greater than $1 - \epsilon$,

$$\mathbb{E}[\mathcal{L}(f(\mathbf{x}), y) \wedge M] \leq \frac{1}{n} \sum_{i=1}^{n} \mathcal{L}(f(\mathbf{x}_i), y_i) \wedge M + \frac{6\widetilde{C}'(\xi)^{\nu/\tilde{d}}}{n^{(\nu/\tilde{d}+1)/2}(\tilde{d}/\nu - 1)^{\nu/\tilde{d}}}$$

$$+ \sqrt{\frac{4M^2 \log(2/\epsilon)}{n}} + \frac{32M \log(2/\epsilon)}{3n}.$$

The rest of the proof is similar to those in the proof of Theorem 3, and we omit the details. $\qquad \square$

*Proof of Corollary 2.* The proof is similar to the one for Corollary 1. We omit the details. $\qquad\square$

*Proof of Theorem 6.* We only highlight the steps where the arguments differ from those for Theorem 3. We let $C'$ be a generic constant that can differ from place to place. Note that for some small enough $\lambda > 0$,

$$P(\max_i B^2(y_i) > \varepsilon) \le \sum_{i=1}^n P(B(y_i) > \sqrt{\varepsilon}) \le nC' \exp(-\lambda\sqrt{\varepsilon}).$$

Setting $\varepsilon = \{\log(nC'/\tau)/\lambda\}^2$, we have $\zeta \le \zeta_0 := \tilde{\alpha}^3 \log(2\tilde{d}\tilde{p})\{\log(nC'/\tau)/\lambda\}^2$ with probability greater than $1 - \tau$. Next, under Assumption 6, we have for some small enough $\lambda > 0$ and large enough $M$,

$$\begin{aligned}
|R(f) - \mathbb{E}[\mathcal{L}(f(\mathbf{x}), y) \wedge M]| &\le \mathbb{E}[(\mathcal{L}(f(\mathbf{x}), y) - M)\mathbf{1}\{\mathcal{L}(f(\mathbf{x}), y) > M\}] \\
&\le \mathbb{E}[G(\mathbf{x}, y)\mathbf{1}\{G(\mathbf{x}, y) > M\}] \\
&\le C' \exp(-\lambda M).
\end{aligned}$$

Moreover,

$$\left| \frac{1}{n}\sum_{i=1}^n \mathcal{L}(f(\mathbf{x}_i), y_i) - \frac{1}{n}\sum_{i=1}^n \mathcal{L}(f(\mathbf{x}_i), y_i) \wedge M \right| \le \frac{1}{n}\sum_{i=1}^n Z_i,$$

where $Z_i = G(\mathbf{x}_i, y_i)\mathbf{1}\{G(\mathbf{x}_i, y_i) > M\}$. Let $\mu_Z = \mathbb{E}[Z_i]$. By Bernstein's inequality, we have for $\varepsilon - \mu_Z > 0$ sufficiently small,

$$\begin{aligned}
P\left( \frac{1}{n}\sum_{i=1}^n Z_i > \varepsilon \right) &= P\left( \frac{1}{n}\sum_{i=1}^n (Z_i - \mu_Z) > \varepsilon - \mu_Z \right) \\
&\le \exp\left( -cn\min\left\{ \frac{|\varepsilon - \mu_Z|}{C'}, \frac{(\varepsilon - \mu_Z)^2}{C'^2} \right\} \right),
\end{aligned}$$

where $c$ is a positive constant. Set $M = k\log(n)$ for some large enough $k$. We have $\mu_Z = O(n^{-\lambda K}) = o(n^{-1})$. Letting $\varepsilon = C'\{\log(1/\eta)/(cn)\}^{1/2} + \mu_Z$ with, we obtain

$$P\left( \frac{1}{n}\sum_{i=1}^n Z_i > \varepsilon \right) \le \eta.$$

Thus, the result follows. $\qquad\square$

*Proof of Corollary 3.* By Theorem 6 and the definition of $\widehat{f}$, we have with probability greater than $1 - \epsilon - \tau - \eta$,

$$\begin{aligned}
R(\widehat{f}) \le &\frac{1}{n}\sum_{i=1}^n \mathcal{L}(f^*(\mathbf{x}_i), y_i) + \frac{144\sqrt{\zeta_0}\{\log(C'n\log(n)/\sqrt{\zeta_0}) \vee 1\}}{n} \\
&+ \frac{C'\log(n)\sqrt{\log(2/\epsilon)}}{\sqrt{n}} + \frac{C'\log(n)\log(2/\epsilon)}{3n} + C'\sqrt{\frac{\log(1/\eta)}{n}},
\end{aligned}$$

By Bernstein's inequality, we have

$$P\left( \left| \frac{1}{n}\sum_{i=1}^n \mathcal{L}(f^*(\mathbf{x}_i), y_i) - R(f^*) \right| > \varepsilon \right) \le \exp\left( -cn\min\left\{ \frac{\varepsilon}{C'}, \frac{\varepsilon^2}{C'^2} \right\} \right).$$

The conclusion follows by setting $\varepsilon = C'\{\log(1/\eta)/(cn)\}^{1/2}$. $\qquad\square$

