# OpenReview forum: "Generalization Bounds and Model Complexity for Kolmogorov–Arnold Networks"
_ICLR.cc/2025/Conference — ICLR 2025 Poster_

### Official Review · Reviewer_geZs · 2024-10-20

**Soundness:** 3
**Presentation:** 3
**Contribution:** 2
**Rating:** 6
**Confidence:** 3

**Summary:**

The paper presents generalization bounds for KANs, where the activation functions are learned from the datasets. The paper considers activation functions that are eighter represented by linear combinations of basis functions or lying in a low-dimensional RKHS. The paper presents generalization bounds based on covering number analysis under some assumptions on the Lipschitzness of the activation functions and the l1-norm on the coefficients. The paper further shows that the analysis can be extended to unbounded loss functions, which can be applied to general regression loss functions. Experimental results are also given to verify the theoretical findings.

**Strengths:**

KANs have found various applications in many scientific and engineering problems. The paper gives solid generalization analysis for KANs. The derived generalization bounds clearly show how the parameters affect the generalization, including the Lipschitzness and l1-norm constraint on the coefficients.

**Weaknesses:**

- The technical contribution seems to be limited. As far as I see, the theoretical analysis follows from existing covering number estimates of fully-connected neural networks [R1]. The idea is to build a recursive relationship of covering number estimates for neural networks with different layers (Proposition 1), which shares similarity to Lemma A.7 in [R1]. Furthermore, Proposition 2 also essentially similar to Lemma 3.2 in R1.


- The paper also gives generalization bounds for learning with unbounded loss functions. However, the results admit some drawbacks. First, the bound in Theorem 3 has a linear dependency $1/\eta$, where $\eta$ is a confidence parameter. This is much worse than the logarithmic dependency. Therefore, the results for unbounded loss functions are not with high probability. Indeed, they are just derived by truncating the prediction followed by Markov's inequality. Second, the bound involves $n^{(1-s)/(2s)}$ and would be much worse than the case with bounded loss function ($n^{-1/2}$). For example, if $s=2$, then the bound in Theorem 3 is of the order of $n^{-1/4}$.





R1: Bartlett, P.L., Foster, D.J. and Telgarsky, M.J., 2017. Spectrally-normalized margin bounds for neural networks. Advances in neural information processing systems, 30.

**Questions:**

- It would be nice if the authors can included a detailed discussion on the technical challenge in the covering number estimates of the KANs. Especially, the authors should clarify the novelty of the theoretical analysis.



- Is it possible to give high-probability generalization bounds for unbounded loss functions.

- There is a lacking discussions of the generalization bounds for KANs with those for fully connected networks. Is there a clear advantage on the generalization behavior by theoretical analysis in the paper?

Minor point:

- $\mathcal{F}$ in Proposition 2 is not defined.

---

> ### Author Response · Authors · 2024-11-25
> **Response to Reviewer geZs: Part I**
>
> *1. The technical contribution seems to be limited. As far as I see, the theoretical analysis follows from existing covering number estimates of fully-connected neural networks [R1]. The idea is to build a recursive relationship of covering number estimates for neural networks with different layers (Proposition 1), which shares similarity to Lemma A.7 in [R1]. Furthermore, Proposition 2 also essentially similar to Lemma 3.2 in [R1].*
>
> **Ans:** While some of the techniques used in our analysis are based on prior work in [R1], our analysis yields several new findings: (i) excess risk bounds concerning unbounded loss functions and (ii) results related to activation functions within low-rank Sobolev spaces. Importantly, our innovation lies in applying these theoretical tools to analyze new types of deep neural network structures that have not been previously studied to the best of our knowledge. Please refer to page 3 of the revision for the discussions.
>
> *2. The paper also gives generalization bounds for learning with unbounded loss functions. However, the results admit some drawbacks. First, the bound in Theorem 3 has a linear dependency $1/\eta$, where $\eta$ is a confidence parameter. This is much worse than the logarithmic dependency. Therefore, the results for unbounded loss functions are not with high probability. Indeed, they are just derived by truncating the prediction followed by Markov's inequality. Second, the bound involves $n^{(1-s)/(2s)}$ and would be much worse than the case with bounded loss function $O(n^{-1/2})$. For example, if $s=2$, then the bound in Theorem 3 is of the order of $n^{-1/4}$.*
>
> **Ans:** We appreciate your thoughtful comments. The current result is due to the relatively weak moment assumption we impose and the Markov inequality used in the proof. Indeed, if we assume that $G(\mathbf{x},y)$ and $B(y)$ both have sub-exponential tails, then by using a more delicate analysis and Bernstein's inequality, we can obtain improved rates and dependence on $\eta$. Specifically, the bound in Theorem 3 can be improved to that with probability greater than $1-\epsilon-\tau-\eta$
>
> \begin{align*}
> R(f)\leq & \frac{1}{n}\sum^{n}_{i=1}\mathcal{L}(f(\mathbf{x}_i),y_i) + \frac{144\sqrt{\zeta_0}\{\log(C n\log(n)/\sqrt{\zeta_0})\vee 1\}}{n}
> +\frac{C\log(n)\sqrt{\log(2/\epsilon)}}{\sqrt{n}}+\frac{C\log(n)\log(2/\epsilon)}{3n} + C\sqrt{\frac{\log(1/\eta)}{n}},
> \end{align*}
>
> where $\zeta_0=\tilde{\alpha}^3\log(2\tilde{d}\tilde{p})(C\log(nC/\tau))^2$ and $C$ is a large enough constant. Note that the dependency $1/\eta$ in the original bound has been improved to $\sqrt{\log(1/\eta)}$ by using the Bernstein inequality and the term $n^{(1-s)/(2s)}$ has been improved to $n^{-1/2}$ due to the sub-exponential tails. Essentially, there is a trade-off between the moment condition and the rate we could achieve. Due to the page limit, we included these results and discussions in the appendix.
>
> *3. It would be nice if the authors can included a detailed discussion on the technical challenge in the covering number estimates of the KANs. Especially, the authors should clarify the novelty of the theoretical analysis.*
>
> **Ans:** Thank you for the comment. Let us highlight the key technical challenges (in addition to those for MLPs) in deriving covering number estimates for KANs. The first challenge involves finding an appropriate application of Maurey's sparsification lemma, specifically in constructing $V_1,\dots,V_N$ in Lemma 1 for KANs (which is not as straightforward as the case of MLPs). The second challenge lies in controlling the effect of $\mathbf{\Psi}(\mathbf{0})$ on the covering numbers. The term $\mathbf{\Psi}(\mathbf{0})$ is similar to the ReLU function evaluated at the bias for MLPs (note that for MLPs, $\mathbf{\Psi}_l(\mathbf{x}) = \sigma(\mathbf{W}_l\mathbf{x} + \mathbf{b}_l)$ and $\mathbf{\Psi}(\mathbf{0}) = \sigma(\mathbf{b}_l)$). Previous analyses like [R1] did not fully address the impact of the bias terms. Our analysis provides a more complete treatment of this aspect.
>
> Although our proof techniques build upon existing tools, we extend beyond previous results in several ways: (i) We showed to what extent the empirical process theory can be adapted to deal with a more general type of network structure (which indeed includes MLPs as a special case); (ii) We establish excess risk bounds for unbounded loss functions (now strengthened with high-probability bounds and improved convergence rates); (iii) We analyze KANs with activation functions lying in low-rank Sobolev spaces. With our new findings, we demonstrate how classical techniques can be adapted to handle more general and flexible network structures that have not been previously analyzed.

---

> ### Author Response · Authors · 2024-11-25
> **Response to Reviewer geZs: Part II**
>
> *4. Is it possible to give high-probability generalization bounds for unbounded loss functions?*
>
> **Ans:** Related to our answer to your second point, we have derived high-probability generalization bounds for unbounded loss functions. Please refer to Section A.2 of the appendix for the new results.
>
> *5. There is a lacking discussions of the generalization bounds for KANs with those for fully connected networks. Is there a clear advantage on the generalization behavior by theoretical analysis in the paper?*
>
> **Ans:** For a fair comparison, let us consider the bound derived in Theorem 2 with $C=0$ and $\max_i B(y_i)$ being finite. Note that $\zeta = \tilde{\alpha}^3 \log(2\tilde{d}\tilde{p}) \max_i B^2(y_i)$ because this setting is closest to the one for MLPs considered in [R1]. Ignoring the smaller order terms, the bound can be expressed as
>
> $$O\left(
> \frac{||\mathbf{X}||R _ {\text{KAN}}}{n} \log(n)\log(2\tilde{d}\tilde{p}) + \sqrt{\frac{\log(1/\epsilon)}{n}}\right),$$
>
> where $R _ {\text{KAN}}=\left(\prod^{L} _ {j=1}\rho _ j\right)\left(\sum^{L} _ {i=1}(B_ic_i)^{2/3}\right)^{3/2}.$ There have been several results in the literature concerning risk bounds or estimation error bounds for MLP or feedforward networks, such as [R1, R3, R5]. The most closely related work is [R1], which shows that with probability greater than $1-\epsilon$, the error bound is of the order
>
> \begin{align*}
> O\left(\frac{||\mathbf{X}||R_{\mathcal{A}}}{\gamma n}\log(n)\log(W)+\sqrt{\frac{\log(1/\epsilon)}{n}}\right),
> \end{align*}
>
> where $R_{\mathcal{A}}$ is a measure of model complexity (similar to $R$ defined above), $\gamma$ is the margin in the ramp loss and $W$ is the network width (similar to $\tilde{d}$ in our case).
>
> When $R$ and $R_{\mathcal{A}}$ are of the same order (denoted by $R\asymp R_{\mathcal{A}}$), the two bounds are also of the same order. This is not surprising, as $R \asymp R _ {\mathcal{A}}$ implies that the two classes have the same order of complexities. On the other hand, if $R = o(R _ {\mathcal{A}})$ and $R _ {\mathcal{A}}/\sqrt{n} \rightarrow +\infty$, then the bound for KANs is of a smaller order than the bound for MLPs. This situation occurs when the underlying true regression function can be more effectively approximated using KANs (e.g., the true regression function can be exactly expressed by a KAN network). As discussed in [R4], KANs may require fewer parameters than MLPs to approximate the same underlying true function because KANs leverage the inherently low-dimensional representation of the true function, whereas MLPs do not. Below, we provide a numerical example to support this claim.
>
> We use the symbol $\phi$ with different subscripts to denote different spline functions (linear combinations of spline basis). We consider the regression settings, $y=f(x)+\epsilon$, where $\epsilon$ is a mean-zero random error, and:
>
> **Example 1.** $f(x)=\sum_{j=0}^{99}\phi_j(x_j)$, which can be exactly represented by a KAN of shape [100, 1];
>
> **Example 2.** $f(x)=\phi_{1,0}(\sum_{j=0}^{99}\phi_{0,0,j}(x_j))+\phi_{1,1}(\sum_{j=0}^{99}\phi_{0,1,j}(x_j))$, which can be exactly represented by a KAN of shape [100, 2, 1].
>
> We employed KAN with shapes [100, 1], [100, 2, 1],  and MLP with shapes [100, 6, 1] and [100, 12, 1] for the two settings, respectively. Since the number of basis splines used in KAN is 6, the widths for the MLP were set so that the two networks have a similar number of parameters. The test errors (mean squared errors) are shown in the following table.
>
> |       |  Example 1   | Example 2 |
> |-------|---------|-----------|
> | KAN   | 2.0765  | 1.1184e-02|
> | MLP   | 10.5546 | 5.8772e-02|
>
> We observe that the test error of the MLP network is approximately five times that of the KAN network. We also tested MLPs with different shapes and observed that neither decreasing nor increasing the widths or depths significantly improved the test error. Although the examples considered are special cases where the regression functions can be exactly represented by a KAN, we point out that the MLP is a special structure of KAN (e.g., when the identity function is included as a basis function and ReLU is used as the activation function). Therefore, the KAN network has a greater capability to reflect the underlying true regression function, and when it does so, we can expect a lower test error. A similar phenomenon has been observed in [R4]; see Figures 3.1-3.3 therein.

---

> > ### Comment · Reviewer_geZs · 2024-11-27
> > **Official comment by Reviewer geZs**
> >
> > I thank the authors for their response to my previous comments and the revision. I have increased the score since the authors give high-probability bounds and recover the bound $O(1/sqrt{n})$ under a sub-exponential tail assumption. I do not give higher scores since several techniques in the analysis use the idea in [R1].

---

### Official Review · Reviewer_xDbU · 2024-10-27

**Soundness:** 3
**Presentation:** 2
**Contribution:** 2
**Rating:** 5
**Confidence:** 2

**Summary:**

The paper studies generalization bounds for Kolmogorov–Arnold Network (KAN) using covering numbers (through Rademacher complexity). The authors further present a quantity $\alpha$ which they dub the network complexity and show empirically this measure qualitatively follows the excess risk.

**Strengths:**

**Originality**. The contribution seems to be novel. To the best of my knowledge, this is the first work to present generalization bounds for KANs.

**Quality**. The theoretical contribution seems to be sound.

**Clarity**. The paper is well organized.

**Significance**. KANs have attracted much attention from the community lately and theoretical contributions would be very welcome.

**Weaknesses:**

**Clarity**. While the flow of the paper is clear, I wish technical results were complemented by an intuitive discussion. Specifically,
- The third point under main contributions claims “The results shed new light on how the performance of KANs is affected by the underlying network structure and the network complexity.” However, this point is never presented or discussed in the paper.
- Empirical results lack discussion completely.

**Quality**. The empirical results should be more comprehensive and include more details.
- Some very particular choices are made for the synthetic target functions, however, they are never motivated or explained.
- The performance of the networks (loss/accuracy rather than excess loss) is never presented. This is important in order to evaluate whether the regime the authors present in their result is an interesting one or not.
- The paper does not include empirical results for the generalization bounds. It would be of great interest to the reader to see whether the bound is tight.
- The tightness of the bound is never discussed.
- I would expect an appendix with details of the experiments. Specifically, the “normalization” procedure for the complexity measure is never explained.

**Significance**. The significance of the result largely depends on the tightness of the bound, but it is never discussed or examined.

*Nits*:
- The appendices are never referenced. I would expect them to be referenced when relevant.
- The text in the figures is barely readable at any reasonable “zoom” level.

**Questions:**

- For the CIFAR-10 and MNIST experiments, why did the authors choose to run the experiments on top of the features of AlexNet instead of the “raw” input? How would that change the results presented in the paper?
- Figure 2 presents the excess loss and the “normalized” complexity measure as a function of the training time, however, it is never stated explicitly how the complexity measure depends on the training time. Am I correct that this dependence comes from the network parameters $B_i$? What are the values of $B_i$ used for the plot? Is it the $\ell_1$ norms of $\mathbf{B}_i$?
- Is it true that the theoretical results do not hold for cross-entropy loss? Can the authors comment on that in light of the use of cross-entropy loss in the experiments?
- Can the authors provide an intuitive explanation for the role played by the value of $\phi$ evaluated at $0$ (and the constant $C$)?

---

> ### Author Response · Authors · 2024-11-25
> **Response to Reviewer xDbU: Part I**
>
> **Clarity** *While the flow of the paper is clear, I wish technical results were complemented by an intuitive discussion. Specifically,*
>
> *1. The third point under main contributions claims ``The results shed new light on how the performance of KANs is affected by the underlying network structure and the network complexity.'' However, this point is never presented or discussed in the paper.*
>
> **Ans:** Thank you for raising this point. We acknowledge that the connection between our theoretical results and practical insights could have been articulated more clearly. Our contribution regarding network structure and complexity is demonstrated through several key aspects of our work:
>
> (i) Our derived generalization bound explicitly includes a complexity term, $\tilde{\alpha}$, which quantifies the structural properties of KANs. This provides a theoretical measure of how choices in network architecture impact learning performance.
>
> (ii) Our simulation results in Section 3 demonstrate a strong empirical correlation between this complexity measure and the observed generalization error. Specifically, we found that the difference between the test loss and training loss is strongly correlated with $\tilde{\alpha}$ in our numerical examples.
>
> (iii) This theoretical-empirical connection suggests a practical design principle: the generalization performance of KANs can be improved by controlling this complexity measure. This insight naturally leads to a potential regularization strategy where the identified complexity term could be used as a regularizer in the training objective.
>
> We have provided a more detailed discussion about this point in the revision.
>
> *2. Empirical results lack discussion completely.*
>
> **Ans:** We have added more details and discussions of the empirical results in Section A.4 of the revision.
>
> **Quality.** *The empirical results should be more comprehensive and include more details.*
>
> *1. Some very particular choices are made for the synthetic target functions, however, they are never motivated or explained.*
>
> **Ans:** Thank you for the comments. The regression functions used in the simulation studies are two examples considered in the original KAN paper [R4]. They serve as representative examples of synthetic functions for the low-dimensional and high-dimensional cases, respectively.
>
> *2. The performance of the networks (loss/accuracy rather than excess loss) is never presented. This is important in order to evaluate whether the regime the authors present in their result is an interesting one or not.*
>
> **Ans:** We have included the figure showing the curves of KAN complexity and test loss in Figure 3 and provided additional details of experiments in Section A.4 of the revision.
>
> *3. The paper does not include empirical results for the generalization bounds. It would be of great interest to the reader to see whether the bound is tight.*
>
> **Ans:** Directly comparing the bound with the generalization error in finite samples may not be particularly meaningful, as the bound depends on some unknown constants, and the coefficients of certain terms in the bound are unlikely to be tight.
> One practical value of our findings is that they help us understand (i) how to quantitatively represent model complexity in KANs and (ii) how model complexity can influence generalization performance. Specifically, model complexity can be represented in a compact form using the scalar:
>
> $$R _ {\text{KAN}} = \left(\prod _ {j=1}^{L}\rho_j\right)\left(\sum _ {i=1}^{L}(B_ic_i)^{2/3}\right)^{3/2}.$$
>
> Our empirical results indicate a strong correlation between excess loss/test loss and $R_{\text{KAN}}$. This finding highlights the importance of developing regularization techniques, whether explicit or implicit (such as early stopping or penalization), to control this complexity measure, which could potentially improve generalization performance. We have included additional numerical results and discussions in Section A.4 of the appendix.

---

> > ### Author Response · Authors · 2024-11-25
> > **Response to Reviewer xDbU: Part II**
> >
> > *4. The tightness of the bound is never discussed.*
> >
> > **Ans:** Please refer to our response to your comment on the significance.
> >
> > *5. I would expect an appendix with details of the experiments. Specifically, the “normalization” procedure for the complexity measure is never explained.*
> >
> > **Ans:** In Figure 2, we display the curves of the KAN complexity as measured by $(\prod^{L} _ {j=1}\rho _ j)^{2/3}\sum^{L} _ {i=1}(B_ic_i)^{2/3}$ and the difference between the test loss and training loss on the same plot. Since the values of these two terms are on different scales, we normalize the complexity so that the two curves meet at the end. Specifically, let $u=(u_1,\dots,u_N)$ be the values of the differences between the test losses and training losses, where $N$ is the number of training epochs, and $v=(v_1,\dots,v_N)$ be the model complexity. Let $v_{\text{max}}=\max(v_1,\dots,v_N)$ and $v_{\text{min}}=\min(v_1,\dots,v_N)$. The normalized complexity is $v'=(v'_1,\dots,v'_N)$ with
> >
> > $$v' _ i=\frac{(v _ i-v _ {\text{min}})u_N}{v_{\text{max}}-v_{\text{min}}}.$$
> >
> > We have provided more details of the experiments, including this definition of the normalization procedure for the complexity measure, in Section A.4 of the revision.
> >
> > **Significance.** *The significance of the result largely depends on the tightness of the bound, but it is never discussed or examined.*
> >
> > **Ans:** We appreciate your comment. To understand the tightness of the bound, let us consider the bound derived in Theorem 2. To simplify the bound, we assume that $C=0$ (i.e., the activation functions have been centered) and $\max _ i B(y_i)$ is finite (which is true for classification problems). Recall from the paper that $\zeta = \tilde{\alpha}^3 \log(2\tilde{d}\tilde{p}) \max_i B^2(y_i)$. Ignoring the smaller order terms, the bound can be expressed as
> >
> > $$O\left(
> > \frac{||\mathbf{X}||R _ {\text{KAN}}}{n} \log(n)\log(2\tilde{d}\tilde{p}) + \sqrt{\frac{\log(1/\epsilon)}{n}}\right),$$
> >
> > where $R _ {\text{KAN}}=(\prod^{L} _ {j=1}\rho_j)(\sum^{L} _ {i=1}(B_ic_i)^{2/3})^{3/2}$ is a measure of the network complexity. We note that $||\mathbf{X}||=O(n)$ when the input dimension is fixed asymptotically. Thus when $R=O((\log(n))^a)$ for some $a>0$, the bound achieves the $O(n^{-1/2})$ rate up to some logarithmic factor. This rate aligns with the best possible rate for the bounds for MLPs discussed in [R1].
> >
> > The bottleneck of the current analysis comes from the use of the Rademacher complexity, which results in a rate of at least $O(n^{-1/2})$. Proving the tightness of the bounds is indeed a challenging task (even for MLPs in some problems). To partly address this concern, we derive a lower bound on the Rademacher complexity of the class induced by KANs; see Section A.3 of the appendix. Moreover, we can partly show the tightness of the bound in Theorem 2 by using the following fact. Suppose $\mathcal{L}$ satisfies that $|\mathcal{L}(f(\mathbf{x}),y)|\leq M$ for any $\mathbf{x},y$ and $f\in\mathcal{M}$. It is known that for any $\delta>0$,
> >
> > $$\sup _ {f\in\mathcal{M}}\left|R(f)-\frac{1}{n}\sum^{n} _ {i=1}\mathcal{L}(f(\mathbf{x} _ i),y _ i)\right|  \geq \frac{\mathcal{R}(\mathcal{M} _ {\mathcal{L}}(\mathbf{X}))}{2} - \frac{\sup _ {f\in\mathcal{M}}|\mathbb{E}[\mathcal{L}(f(\mathbf{x}),y)]|}{\sqrt{n}} -\delta
> > $$
> >
> > with probability at least $1-\exp(-n\delta^2/(2M^2)).$ When $\mathcal{R}(\mathcal{M} _ {\mathcal{L}}(\mathbf{X}))=O(n^{-1/2})$, the bound in Theorems 2-3 are nearly tight (up to some logarithmic factors) when $R$ does not grow too fast. However, showing a lower bound for
> >
> > $R(\widehat{f})-\frac{1}{n}\sum^{n}_{i=1}\mathcal{L}(\widehat{f}(\mathbf{x}_i),y_i)$
> >
> > is quite challenging and requires a more delicate analysis. We shall investigate this interesting problem in our future research.
> >
> > **Nits:** (i) The appendices are never referenced. I would expect them to be referenced when relevant. (ii) The text in the figures is barely readable at any reasonable “zoom” level.
> >
> > **Ans:** Thanks for the comments. We have fixed these issues in the revision.

---

> > > ### Author Response · Authors · 2024-11-25
> > > **Response to Reviewer xDbU: Part III**
> > >
> > > **Questions**
> > >
> > > *1. For the CIFAR-10 and MNIST experiments, why did the authors choose to run the experiments on top of the features of AlexNet instead of the ``raw'' input? How would that change the results presented in the paper?*
> > >
> > > **Ans:** Since the purpose of our simulation studies is to demonstrate the practical relevance of the generalization bound derived in Section 2, we use the pre-trained features instead of raw inputs so that the KAN structure can be directly applied. Using the raw input would require a more careful design of the network structure, which is beyond the scope of the current work.
> > > We have noted that the convolutional KAN has been proposed by [R2]. Calculating the model complexity for convolutional KANs requires more careful consideration, which we leave as future work.
> > >
> > > *2. Figure 2 presents the excess loss and the “normalized” complexity measure as a function of the training time, however, it is never stated explicitly how the complexity measure depends on the training time. Am I correct that this dependence comes from the network parameters $B_i$? What are the values of $B_i$ used for the plot? Is it the $l_1$ norms of $\mathbf{B}_i$?*
> > >
> > > **Ans:** We use $(\prod^{L} _ {i=1}\rho_i)^{2/3}\sum^{L} _ {i=1}(B_ic_i)^{2/3}$ as the measure of complexity. During the training process, the weights of the basis functions (denoted by $\mathbf{B}_i$) change, leading to corresponding changes in $\rho_i$, which depends on the spectral norm of the weights, and $B_i$, which represents the $l_1$ norm of the weights.
> > >
> > > *3. Is it true that the theoretical results do not hold for cross-entropy loss? Can the authors comment on that in light of the use of cross-entropy loss in the experiments?*
> > >
> > > **Ans:** Our theory applies to the cross-entropy loss, as it permits the loss to be unbounded; see, e.g., Theorems 3 and 5. We have taken the cross-entropy loss into account in our numerical studies. Specifically, in Figure 2, the loss functions for (iii), (iv), MNIST, and CIFAR10 are chosen to be the cross-entropy loss. The cross-entropy loss is smooth and differentiable, making it a commonly used loss function for classification problems.
> > >
> > > *4. Can the authors provide an intuitive explanation for the role played by the value of $\phi$ evaluated at 0 (and the constant $C$)?*
> > >
> > > **Ans:** To clarify the role of $\mathbf{\Psi}_l(\mathbf{0})$, we note that MLPs can be viewed as a specific type of network structure considered in this paper, where $\mathbf{\Psi}_l(\mathbf{x}) = \sigma(\mathbf{W}_l\mathbf{x} + \mathbf{b}_l)$. In this case, $\mathbf{\Psi}(\mathbf{0}) = \sigma(\mathbf{b}_l)$. This means that the value of $\mathbf{\Psi}_l$ evaluated at zero is analogous to the ReLU activation function evaluated at the bias in MLPs. The constant $C$ quantifies the magnitude of $\max_l ||\mathbf{\Psi}_l(\mathbf{0})||_2$. It appears in the upper bound of the term $||\mathbf{\Psi} _ l \circ \mathbf{\Psi} _ {l-1} \circ \cdots \circ \mathbf{\Psi} _ 1(\mathbf{X})||_2$ within the proof of Theorem 1.

---

> > > > ### Comment · Reviewer_xDbU · 2024-11-25
> > > >
> > > > I thank the authors for taking the time to answer my comments and questions. The authors have fully answered my questions and have clarified many of my concerns, but have not fully resolved them.
> > > >
> > > > - Regarding *Quality>3*. The answer suggests they cannot plot their bounds even for cases (i)-(iv) in the manuscript. Is that true?
> > > > - At the risk of misinterpreting the authors, it seems they claim the major contribution of the paper is conceptual (the complexity measure) rather than the generalization bounds themselves, which they believe to be completely impractical. While I agree the conceptual contribution of a complexity measure can be of equal importance the overwhelming majority of the paper is dedicated to presenting generalization bounds instead.
> > > > - The authors suggest using the complexity measure for regularization. Can they support that with experiments?
> > > > -  The new Fig. 3 shows a rather abnormal pattern. It seems that in (iv) the test loss monotonically increases; can the authors address that?
> > > > - On the same topic, the performance of the networks on CIFAR and MNIST is remarkably poor, suggesting that the regime investigated in the experiments is not the natural one to examine. Can the authors comment on that?
> > > >
> > > > I am open to any clarification, correction, or answer on these matters.

---

> > > > > ### Author Response · Authors · 2024-11-29
> > > > > **Response to Reviewer xDbU**
> > > > >
> > > > > Thank you for the additional comments. Let us provide point-by-point replies below.
> > > > >
> > > > > *Regarding Quality>3. The answer suggests they cannot plot their bounds even for cases (i)-(iv) in the manuscript. Is that true?*
> > > > >
> > > > > **Ans:** We can plot these bounds, but doing so does not necessarily indicate whether the bounds are tight with respect to the convergence rate. This is because the bounds involve constant factors, and determining their best possible values is often quite challenging as it may require knowledge of the data distribution (which is unknown to us in reality).
> > > > >
> > > > > Note that the leading term in the bound for $R(f)-\frac{1}{n}\sum^{n}_{i=1}\mathcal{L}(f(\mathbf{x} _ i),y_i)$, as provided in Theorem 2, is
> > > > >
> > > > > \begin{align*}
> > > > > \frac{144\sqrt{\zeta}\{\log(nM/(3\sqrt{\zeta}))\vee 1\}}{n}
> > > > > \end{align*}
> > > > >
> > > > > The variation of this term is determined by $\zeta$ (for a fixed sample size $n$) and consequently by the complexity measure. Our experimental results show that the excess losses follow a similar trend to that of the complexity measure as the training process progresses. Thus, we believe that our complexity measure effectively represents the generalization performance of a KAN network. Our unreported simulations indicate that the generalization bounds follow a pattern similar to that of the complexity measures. However, these bounds do not offer sufficiently precise numerical predictions for the actual excess losses in our simulations, likely due to the non-tight constant scaling factor.
> > > > >
> > > > > *At the risk of misinterpreting the authors, it seems they claim the major contribution of the paper is conceptual (the complexity measure) rather than the generalization bounds themselves, which they believe to be completely impractical. While I agree the conceptual contribution of a complexity measure can be of equal importance the overwhelming majority of the paper is dedicated to presenting generalization bounds instead.*
> > > > >
> > > > > **Ans:** The main contribution of this paper is the theory of the generalization bounds, their implications, and the techniques used to derive these results. It is summarized in Section 1.1. From a practical standpoint, these generalization bounds are valuable for several reasons:  (i) They help us understand why simpler models (e.g., as quantified by the complexity measure) often generalize better. (ii) They provide insights into how much data is necessary to reach desired performance levels. (iii) They help explain why techniques like early stopping, dropout, and other regularization methods are effective. For example, as illustrated in Figure 3, a rapid increase in complexity can lead to poor generalization performance. However, it is important to note that the generalization bounds may not be precise/sharp enough to serve as direct numerical predictors. In our opinion, their primary value lies in the qualitative insights they offer regarding the relationships between key factors (e.g., sample size, input dimension, network width, number of layers, network parameters, and choice of basis functions) that influence generalization. The complexity measure is a practically useful byproduct from these generalization bounds.
> > > > >
> > > > > *The authors suggest using the complexity measure for regularization. Can they support that with experiments?*
> > > > >
> > > > > **Ans:** Thank you for your suggestion. We have added additional numerical results for KANs trained with regularization (i.e., dropout) in Section A.4 of the appendix (refer to Figure 4 for details).
> > > > >
> > > > > *The new Fig. 3 shows a rather abnormal pattern. It seems that in (iv) the test loss monotonically increases; can the authors address that? On the same topic, the performance of the networks on CIFAR and MNIST is remarkably poor, suggesting that the regime investigated in the experiments is not the natural one to examine. Can the authors comment on that?*
> > > > >
> > > > > **Ans:** We observe these patterns because we analyze a network structure that is overparameterized and not using any regularization. Our aim is to understand the relationship between test error and model complexity. In small-scale models, overfitting does not occur, making it hard to fully grasp how complexity affects generalization performance. Following your suggestion, we added numerical results for KANs trained with regularization (i.e., dropout) in Section A.4 of the appendix (refer to Figure 4 for details). Overall, with regularization, KANs deliver much better performances. In this case, the excess loss is again tightly correlated with the complexity measure.

---

> > > > > > ### Comment · Reviewer_xDbU · 2024-12-01
> > > > > >
> > > > > > I thank the authors for their comprehensive response.
> > > > > >
> > > > > > The manuscript derives the first generalization bounds for KANs (to the best of my knowledge), and identifies a quantity which they dub complexity measure for KANs. In their experiments section they attempt to establish this measure is indicative of the real-world behavior of KANs.
> > > > > >
> > > > > > I find the experiments to be under-discussed and insufficient to support their claim. The regime of the experiment is not the one normally considered in the community, namely the one where neural networks are able to perform and generalize reasonably well. While the authors noted their networks are over-parametrized, this does not usually constitute an explanation for such bad generalization. In fact, the manuscript cites the NTK (Jacot et al. 2018); the rich literature on double descent, and the infinite width limit of neural networks has shown that over-parametrized networks do generalize. A discussion of this result is required.  I note that I am not aware of any result parallel to the lazy limit (NTK / NNGP) for KANs so these results do not have to hold in this case, which only makes the discussion more relevant.
> > > > > >
> > > > > > While the paper has significant theoretical contributions, the fact the experimental results contradict common expectations but lack discussion prevents me from recommending the paper for ICLR. I keep my score as is.

---

> ### Author Response · Authors · 2024-12-01
> **Response to Reviewer xDbU**
>
> Thank you for your feedback. The current numerical experiments serve two goals: (i) they provide a cautionary note about training KANs in the over-parameterized regime without regularization, highlighting the necessity of some form of regularization to prevent overfitting; (ii) they demonstrate the relevance of network complexity in predicting a network's generalization performance for both unregularized and regularized KANs (see Figure 4). The newly added numerical results indicate that with appropriate regularization, KANs can achieve significantly improved performance in terms of test loss. In view of Figure 4, the performance of regularized KANs is aligned more with the "common expectations." We will continue investigating regularized KANs and report any interesting findings along this line.
>
> It is true that neural networks can perform well even when over-parameterized, but this does not imply that they can do so without any form of regularization. With proper regularization, over-parameterized networks can generalize well and may experience the so-called double descent phenomenon. This is also evident in simpler models, such as ridge/ridgeless high-dimensional least squares regression, as shown in the work of Hastie et al. (2020, https://arxiv.org/pdf/1903.08560), where the underlying estimator is properly regularized (for example, they studied interpolated estimators with the minimum $l_2$ norm). The NTK provides an alternative theory that connects DNNs trained through gradient descent with kernel methods, relating the convergence of DNN training to the positive definiteness of the limiting NTK. Their theory also supports the use of some form of regularization, such as early stopping, which is explicitly mentioned in their work. Our theory and numerical results do not contradict the existing findings and "common expectations" for DNNs in this regard; rather, they provide insights into how model complexity affects generalization and advocate for the use of regularization in KANs.
>
> As KANs are a relatively new type of network structure, there are still many open questions regarding their potential and behavior under different settings. For instance, do KANs exhibit the double descent phenomenon? As you pointed out, it remains unclear whether one can establish results parallel to those of NTK/NNGP. What is the most effective way to regularize KANs? We view our work as an initial step in understanding KAN generalization, and we believe the above questions are all worthwhile research problems that deserve further investigation.

---

### Official Review · Reviewer_u9e2 · 2024-11-01

**Soundness:** 2
**Presentation:** 3
**Contribution:** 2
**Rating:** 6
**Confidence:** 4

**Summary:**

This paper investigates the generalization ability of Kolmogorov-Arnold Networks (KAN). Multiple generalization bounds are provided for both bounded and unbounded loss functions, which may help understanding how the network complexity impacts the generalization behavior of KANs. The results are further extended to the cases where activation functions belong to RKHS, and empirical results are given to demonstrate the relationship between performance and network complexity.

**Strengths:**

This paper provides multiple generalization bounds for both bounded and unbounded loss functions, which may help in understanding how the network complexity impacts the generalization behavior of KANs.

**Weaknesses:**

* It is consensus that traditional complexity measures (e.g. Rademacher complexity or covering numbers) are not sufficient to explain generalization for deep networks. As the analysis of this paper heavily relies on these metrics, the derived bounds are likely vacuous in practice.
* The theoretical analysis relies on tons of strong assumptions (e.g. Lipschitzness, boundedness) for almost every single part of the network (input, function space, loss function). To what extent these assumptions can be met in practice needs more clarification.

**Questions:**

* Could the authors provide some empirical results on the comparison between the bounds and the actual generalization error? This would help convince me that these bounds are indeed non-vacuous in practice.

---

> ### Author Response · Authors · 2024-11-25
> **Response to Reviewer u9e2: Part I**
>
> *1. It is consensus that traditional complexity measures (e.g. Rademacher complexity or covering numbers) are not sufficient to explain generalization for deep networks. As the analysis of this paper heavily relies on these metrics, the derived bounds are likely vacuous in practice.*
>
> **Ans:** We appreciate your thoughtful comment. While we acknowledge that traditional complexity measures alone may not fully explain the generalization behavior of deep neural networks - we believe there remains value in theoretical analyses using these classical tools for several reasons:
>
> 1. Foundational Understanding: Classical complexity measures offer a rigorous mathematical framework that can provide insights into the fundamental properties of neural networks. In some instances, the bounds derived using these measures and tools have been shown to achieve optimal rates, such as the minimax rate.
>
> 2. Complementary Perspective: Our analysis using traditional metrics complements rather than contradicts newer approaches. Recent works have shown that combining classical tools with neural network-specific considerations can lead to meaningful results, e.g., [R1, R3, R5].
>
> 3. Complexity Measures: The complexity measures involved in the bounds (i.e., the quantity $\tilde{\alpha}$) shed some light on how the generalization error is affected by the network structure, which might motivate ways to regularize KANs during the training process. Indeed, our numerical results suggest that these measures are strongly correlated with the difference between the test loss and training loss and thus could be practically relevant. Similar findings for MLPs were also reported in [R1].
>
> In the revision, we have added discussions acknowledging the known limitations of traditional complexity measures in the analysis of deep neural networks and clarified how our analysis fits into the broader landscape of generalization theory; see page 3 of the revision.
>
> *2. The theoretical analysis relies on tons of strong assumptions (e.g. Lipschitzness, boundedness) for almost every single part of the network (input, function space, loss function). To what extent these assumptions can be met in practice needs more clarification.*
>
> **Ans:** Thank you for the comment. Below, we will clarify the practical relevance and applicability of these assumptions.
>
> 1. Lipschitzness: Lipschitz continuity is a widely used assumption in nonparametric statistics. Function classes that exhibit Lipschitz continuity include: (i) differentiable functions with bounded derivatives, (ii) convex functions, (iii) monotonic functions, and (iv) piecewise linear functions. Notably, functions that belong to the Hölder space, Sobolev space, or the space of continuous and bounded variation are also Lipschitz continuous.
>
> 2. Boundedness: First, the assumption of input boundedness (i.e., $||\mathbf{X}||\leq D$) is not unreasonable, as real-world data typically exists within a compact domain. For example, normalized image pixels fall within the range $[0,1]$, and standardized features in tabular data adhere to similar constraints. We can relax this assumption by only requiring that $||\mathbf{X}|| \leq D$ holds with high probability, specifically that $P(||\mathbf{X}||\leq D) \geq 1-\epsilon$ for a small value of $\epsilon$, where the constant $D$ is allowed to grow with $n$. Bounded functional space is a natural requirement in empirical process theory; without it, the complexity of the functional space (as measured by covering numbers or Rademacher complexity) is infinity, making technical analysis infeasible. While the boundedness condition on the loss is valid for certain cases, such as non-convex least squares loss, Ramp loss, and Ramp $\epsilon$-insensitive loss, there are also numerous instances where the loss function is unbounded. We acknowledge this limitation and derive the generalization error bounds for unbounded loss functions in the paper; see, for example, Theorems 3 and 5.

---

> > ### Author Response · Authors · 2024-11-25
> > **Response to Reviewer u9e2: Part II**
> >
> > *3. Could the authors provide some empirical results on the comparison between the bounds and the actual generalization error? This would help convince me that these bounds are indeed non-vacuous in practice.*
> >
> > **Ans:** Directly comparing the bounds with the generalization errors in finite samples may not be particularly meaningful, as the bounds depend on some unknown constants, and the coefficients of certain terms in the bound are unlikely to be tight.
> > One practical value of our findings is that they help us understand (i) how to quantitatively represent model complexity in KANs and (ii) how model complexity can influence generalization performance. Specifically, model complexity can be represented in a compact form using the scalar:
> >
> > $R _ {\text{KAN}} = \left(\prod _ {j=1}^{L}\rho _ j\right)\left(\sum _ {i=1}^{L}(B_ic_i)^{2/3}\right)^{3/2}$.
> >
> > Our empirical results indicate a strong correlation between excess loss/test loss and $R _ {\text{KAN}}$. This finding highlights the importance of developing regularization techniques, whether explicit or implicit (such as early stopping or penalization), to control this complexity measure, which could potentially improve generalization performance. We have included additional numerical results and discussions in Section A.4 of the appendix.

---

> > > ### Comment · Reviewer_u9e2 · 2024-11-26
> > >
> > > I would like to thank the authors for their response. After reading the rebuttal, I would like to increase my score to 6 to acknowledge the efforts of the authors, and not higher because I still believe the adopted traditional complexity measures are insufficient to fully explain the generalization of deep networks.

---

### Official Review · Reviewer_mrKs · 2024-11-04

**Soundness:** 3
**Presentation:** 2
**Contribution:** 3
**Rating:** 6
**Confidence:** 3

**Summary:**

This paper explores the generalization capability of Kolmogorov-Arnold Networks (KANs). Specifically, the authors estimated the covering number under certain regularity assumptions and derived the generalization gaps (Theorems 2 and 3) and the excess risk bound (Corollary 1). Additionally, they examined the case where the activation function belongs to a Reproducing Kernel Hilbert Space (RKHS) with a low-rank structure, proving the generalization gaps (Theorems 4 and 5) and the excess risk bound (Corollary 2) that leverage this structure. Experiments validate the relevance of the theoretical bounds to the observed excess risk.

**Strengths:**

KANs, inspired by the Kolmogorov-Arnold representation theorem, have gained attention for potentially efficient parameterization compared to MLPs. However, their statistical performance remains unclear, despite numerous attempts to develop more sophisticated structures and applications. To the best of my knowledge, this is the first work to derive generalization bounds for KANs.

**Weaknesses:**

- Assumption 1 presumes the uniform boundedness of the dataset. Typically, this bound scales with $\sqrt{n}$, which could degrade the derived bounds.

- Although KANs are expected to demonstrate superior approximation capability due to their inherent nature, there is no comparison with other models such as MLPs.

**Questions:**

- Based on the obtained theory, is it possible to identify a problem setting where KANs outperform MLPs?

- How tight are the obtained results? Is it feasible to derive lower bounds on generalization gaps and excess risk bounds in a simple setting?

- Can you discuss the approximation (representation) capabilities of KANs for certain function classes? This type of approximation analysis is often necessary for fully understanding the statistical performance of the model.

---

> ### Author Response · Authors · 2024-11-25
> **Response to Reviewer mrKs: Part I**
>
> *1. Assumption 1 presumes the uniform boundedness of the dataset. Typically, this bound scales with $\sqrt{n}$, which could degrade the derived bounds.*
>
> **Ans:** Thanks for the comment. It is true that the bound $D$ generally scales with $\sqrt{n}$. However, this scaling does not degrade the derived bounds. For instance, let us consider the bound derived in Theorem 2. The constant $D$ appears in this bound through $\zeta = \tilde{\alpha}^3 \log(2\tilde{d}\tilde{p})\max _ i B^2(y _ i)$. We note that $\tilde{\alpha}^3$ is proportional to $D^2$, which is of the order $O(n)$ when the input dimension is fixed asymptotically. Therefore, neglecting some smaller order terms, the bound can be expressed as follows:
>
> $$O\left(
> \frac{R _ {\text{KAN}}}{\sqrt{n}}\log(n)\log(2\tilde{d}\tilde{p}) + \sqrt{\frac{\log(1/\epsilon)}{n}}\right),$$
>
> where $R _ {\text{KAN}}=(\prod^{L} _ {j=1}\rho _ j)(\sum^{L} _ {i=1}(B_ic_i)^{2/3})^{3/2}$ is a measure of the network complexity. When $R=O((\log(n))^a)$ for some $a>0$, we achieve the $O(n^{-1/2})$ rate up to some logarithmic factor.
> Indeed, a similar dependence on $D$ appears in the bound for MLPs; see Theorem 1.1 of [R1]. We can relax this assumption by only requiring that $||\mathbf{X}|| \leq D$ holds with high probability, specifically that $P(||\mathbf{X}||\leq D) \geq 1-\epsilon$ for a small value of $\epsilon$.
>
> *2. Although KANs are expected to demonstrate superior approximation capability due to their inherent nature, there is no comparison with other models, such as MLPs. Based on the obtained theory, is it possible to identify a problem setting where KANs outperform MLPs?*
>
> **Ans:** Thank you for the helpful suggestion. To address this, let us begin by making a theoretical comparison between the bounds we derived and those for MLPs. Based on this comparison, we will then present a numerical example to demonstrate that KANs can outperform MLPs in certain scenarios. It is important to emphasize that our primary aim is not merely to advocate for the use of KANs but to gain a deeper understanding of the potential advantages and disadvantages of this new type of network structure from a theoretical perspective.
>
> To make a fair comparison, we again consider the bound derived in Theorem 2 with $C=0$ and $\max_i B(y_i)$ being finite because this setting is closest to the one for MLPs considered in \cite{bartlett2017spectrally}. Recall from the paper that $\zeta = \tilde{\alpha}^3 \log(2\tilde{d}\tilde{p}) \max _ i B^2(y _ i)$. Ignoring the smaller order terms, the bound can be expressed as
>
> $$O\left(
> \frac{||\mathbf{X}||R_{\text{KAN}}}{n} \log(n)\log(2\tilde{d}\tilde{p}) + \sqrt{\frac{\log(1/\epsilon)}{n}}\right).$$
>
> There have been several recent results in the literature concerning risk bounds or estimation error bounds for MLPs, such as [R1, R3, R5]. The most closely related work is [R1], which shows that with probability greater than $1-\epsilon$, the error bound is of the order
>
> \begin{align*}
> O\left(\frac{||\mathbf{X}||R_{\mathcal{A}}}{\gamma n}\log(n)\log(W)+\sqrt{\frac{\log(1/\epsilon)}{n}}\right),
> \end{align*}
>
> where $R_{\mathcal{A}}$ is a measure of model complexity (similar to $R$ defined above), $\gamma$ is the margin in the ramp loss and $W$ is the network width (similar to $\tilde{d}$ in our case).
>
> When $R$ and $R_{\mathcal{A}}$ are of the same order (denoted by $R\asymp R_{\mathcal{A}}$), the two bounds are also of the same order. This is not surprising, as $R \asymp R _ {\mathcal{A}}$ implies that the two classes have the same order of complexities.
> On the other hand, if $R = o(R _ {\mathcal{A}})$ and $R _ {\mathcal{A}}/\sqrt{n} \rightarrow +\infty$, then the bound for KANs is of a smaller order than the bound for MLPs. This situation occurs when the underlying true regression function can be more effectively approximated using KANs (e.g., the true regression function can be exactly expressed by a KAN network). As discussed in [R4], KANs may require fewer parameters than MLPs to approximate the same underlying true function because KANs leverage the inherently low-dimensional representation of the true function, whereas MLPs do not. Below, we provide a numerical example to support this claim.
>
> We use the symbol $\phi$ with different subscripts to denote different spline functions (linear combinations of spline basis). We consider the regression settings, $y=f(x)+\epsilon$, where $\epsilon$ is a mean-zero random error, and:
>
> **Example 1.** $f(x)=\sum_{j=0}^{99}\phi_j(x_j)$, which can be exactly represented by a KAN of shape [100, 1];
>
> **Example 2.** $f(x)=\phi_{1,0}(\sum_{j=0}^{99}\phi_{0,0,j}(x_j))+\phi_{1,1}(\sum_{j=0}^{99}\phi_{0,1,j}(x_j))$, which can be exactly represented by a KAN of shape [100, 2, 1].

---

> ### Author Response · Authors · 2024-11-25
> **Response to Reviewer mrKs: Part II**
>
> We employed KAN with shapes [100, 1], [100, 2, 1],  and MLP with shapes [100, 6, 1] and [100, 12, 1] for the two settings, respectively. Since the number of basis splines used in KAN is 6, the widths for the MLP were set so that the two networks have a similar number of parameters. The test errors (mean squared errors) are shown in the following table.
>
> |       |  Example 1   | Example 2 |
> |-------|---------|-----------|
> | KAN   | 2.0765  | 1.1184e-02|
> | MLP   | 10.5546 | 5.8772e-02|
>
> We observe that the test error of the MLP network is approximately five times that of the KAN network. We also tested MLPs with different shapes and observed that neither decreasing nor increasing the widths or depths significantly improved the test error. Although the examples considered are special cases where the regression functions can be exactly represented by a KAN, we point out that the MLP is a special structure of KAN (e.g., when the identity function is included as a basis function and ReLU is used as the activation function). Therefore, the KAN network has a greater capability to reflect the underlying true regression function, and when it does so, we can expect a lower test error. A similar phenomenon has been observed in [R4]; see Figures 3.1-3.3 therein.
>
> *3. How tight are the obtained results? Is it feasible to derive lower bounds on generalization gaps and excess risk bounds in a simple setting?*
>
> **Ans:** Thank you for your comment. As we mentioned in response to your first comment, our derived bound can achieve a rate of $O(n^{-1/2})$, albeit with some logarithmic factors. This rate aligns with the best possible rate for the bounds for MLPs discussed in [R1]. The bottleneck of the current analysis comes from the use of the Rademacher complexity, which results in a rate of at least $O(n^{-1/2})$. Proving the tightness of the bounds is indeed a challenging task (even for MLPs in some problems). To partly address this concern, we derive a lower bound on the Rademacher complexity of the class induced by KANs; see Section A.3 of the appendix. Moreover, we can partly show the tightness of the bound in Theorem 2 by using the following fact. Suppose $\mathcal{L}$ satisfies that $|\mathcal{L}(f(\mathbf{x}),y)|\leq M$ for any $\mathbf{x},y$ and $f\in\mathcal{M}$. It is known that for any $\delta>0$,
>
> $$\sup _ {f\in\mathcal{M}}\left|R(f)-\frac{1}{n}\sum^{n} _ {i=1}\mathcal{L}(f(\mathbf{x} _ i),y _ i)\right|  \geq \frac{\mathcal{R}(\mathcal{M} _ {\mathcal{L}}(\mathbf{X}))}{2} - \frac{\sup _ {f\in\mathcal{M}}|\mathbb{E}[\mathcal{L}(f(\mathbf{x}),y)]|}{\sqrt{n}} -\delta
> $$
>
> with probability at least $1-\exp(-n\delta^2/(2M^2)).$ When $\mathcal{R}(\mathcal{M} _ {\mathcal{L}}(\mathbf{X}))=O(n^{-1/2})$, the bound in Theorems 2-3 are nearly tight (up to some logarithmic factors) when $R$ does not grow too fast. However, showing a lower bound for
>
> $R(\widehat{f})-\frac{1}{n}\sum^{n}_{i=1}\mathcal{L}(\widehat{f}(\mathbf{x}_i),y_i)$
>
> is quite challenging and requires a more delicate analysis. We shall investigate this interesting problem in our future research.
>
> *4. Can you discuss the approximation (representation) capabilities of KANs for certain function classes? This type of approximation analysis is often necessary for fully understanding the statistical performance of the model.*
>
> **Ans:** The approximation capabilities of KANs generally depend on the form of the true underlying function and the choice of activation functions in the KAN network class. Suppose the underlying true function is defined on $[0,1]^n$ and admits the representation
>
> \begin{align*}
> f _ 0(\cdot)=\mathbf{\Phi} _ L\circ \mathbf{\Phi} _ {L-1}\circ \cdots \circ \mathbf{\Phi} _ 1(\cdot),
> \end{align*}
>
> where $\mathbf{\Phi} _ l=(\phi _ {l,i,j})$ with each $\phi _ {l,i,j}$ being $(k + 1)$ times continuously differentiable. Then, there exists a KAN network with the B-spline functions $\psi _ {l,i,j}$ with grid size $G$ such that
>
> \begin{align*}
> ||f _ 0(\mathbf{x}) - (\mathbf{\Psi} _ {L} \circ \mathbf{\Psi} _ {L-1} \circ \cdots \circ \mathbf{\Psi} _ 1)\mathbf{x} || _ {C^m} \leq c G^{-k-1+m}.
> \end{align*}
>
> where $||g|| _ {C^m} = \max_{|\beta| \leq m} \sup_{\mathbf{x} \in [0,1]^n} |D^\beta g(\mathbf{x})|$ and $c$ is some constant depending on $f_0$. A recent work [R6] shows that MLP with ReLU$^k$ activation functions (i.e., $\sigma(x)=\max(0,x)^k$) can be exactly represented using a KAN with degree $k$ splines. As a result, any function that can be approximated by MLP with ReLU$^k$ activation functions can also be approximated by KANs.

---

> > ### Comment · Reviewer_mrKs · 2024-12-02
> >
> > Thank you for the detailed response. If the authors could demonstrate the superiority of KAN by comparing the upper and lower bounds of the expected risks achieved by KAN and MLP, it would further strengthen the contribution of the paper. That said, the paper is commendable as it provides the first generalization bound for KAN, so I would retain the current evaluation.

---

### Official Review · Reviewer_vaj9 · 2024-11-08

**Soundness:** 3
**Presentation:** 3
**Contribution:** 3
**Rating:** 8
**Confidence:** 4

**Summary:**

This work addresses a gap in the current research on Kolmogorov–Arnold Networks (KANs) by (i) quantifying the complexity of KANs, and (ii) establishing high probability upper bounds of excess risk in regression task. Specifically, this work studies two special classes of KAN: (i) KAN with activation functions represented by basis functions, and (ii) KAN with activation functions on the RKHS induced by Matern kernel.

**Strengths:**

- I found the paper well-written, has a good flow, and I believe it presents very solid theoretical results.
- The theoretical results are novel as they present the first generalization error bound and complexity estimate of KAN.
- Frankly, KAN is an "interesting" yet "contentious" topic in the machine learning community ---   "interesting" in the sense that it has caught a lot of attentions and discussions in the community; "contentious" in the sense that it hasn’t achieved broad acceptance or visibility, especially since all the cited KAN-related papers (including the very first KAN paper (Liu et al. 2024)) have not been published in major machine learning venues. I believe this paper contribute to the theoretical understanding of KAN by estimating the excess risk in regression setting of two special classes of KAN. Although the current results are insufficient to conclude whether KAN achieves better generalization performance than traditional MLPs, they provide valuable insights that can encourage the community to further investigate the potential advantages and impact of KAN.

**Weaknesses:**

-  I find the purpose of the numerical experiments in Section 3 unclear. Are these experiments used to corroborate the complexity result in Theorem 1? If so,  are the activation functions in the implemented KAN chosen to be linear combination of basis functions?
- Can the authors comment further on the convergence rate of excess risk established in Corollary 1 and Corollary 2? What does the parameter $s$ represent? Also, how do these convergence rates compare to that of traditional MLP (say, ReLU feed-forward neural network)? Do we observe any advantage in the generalization of regression by using KAN?
- It appears that Section 2.3 only considers RKHS induced by Matern kernel, instead of a general low-rank RKHS. Because of the equivalence between this specific RKHS and Sobolev space, would it be a better idea to change the title of Section 2.3 to ``Activation functions on Sobolev space"?

**Questions:**

See Weaknesses

---

> ### Author Response · Authors · 2024-11-24
> **Response to reviewer Reviewer vaj9: Part I**
>
> *1. I find the purpose of the numerical experiments in Section 3 unclear. Are these experiments used to corroborate the complexity result in Theorem 1? If so, are the activation functions in the implemented KAN chosen to be linear combination of basis functions?*
>
> **Ans:** The primary objective of the numerical experiments in Section 3 is to showcase the generalization performance of KANs and to analyze how the complexity measure $\tilde{\alpha}$, which appears in the derived bounds, impacts KANs' performance. The activation functions in the implemented KANs are linear combinations of basis functions, including SiLU and Spline basis, as considered by [R4]. We have provided more details in Section A.4 of the revision.
>
> *2. Can the authors comment further on the convergence rate of excess risk established in Corollary 1 and Corollary 2? What does the parameter represent? Also, how do these convergence rates compare to that of traditional MLP (say, ReLU feed-forward neural network)? Do we observe any advantage in the generalization of regression by using KAN?*
>
> **Ans:**  For clarity, we will organize our response into three sections.
>
> **Interpretation of the rate in Corollary 1.** Consider the upper bound on the excess risk in Corollary 1. We first explain what the parameters represent in this bound: (i) $\zeta_0$ is a parameter that quantifies the impact of the KAN network structure on the excess risk. It depends on the model complexity measure $\tilde{\alpha}$, the network width $\tilde{d}$ and the (maximum) number of basis functions $\tilde{p}$ involved in the activation functions; (ii) $\epsilon,\tau$ and $\eta$ determines a trade-off between the bound and the probability of the event that this bound holds. To understand this bound and obtain a more explicit rate, let us ignore the logarithmic terms, the smaller order terms, and the dependence on $\epsilon,\tau$, and $\eta$. The order of the upper bound can be simplified to
>
> \begin{align*}
> O\left(\frac{\sqrt{\zeta_0}}{n}+\frac{1}{n^{(s-1)/(2s)}}\right),
> \end{align*}
>
> where up to a logarithmic factor, $\zeta_0$ is of the order $\tilde{\alpha}^3 n^{2/s'}$ with $\tilde{\alpha}^3$ being proportional to $D^2$. Recall that $D$ is a bound of $||\mathbf{X}|| _ 2$, which is of the order $O(\sqrt{n})$ when the input dimension is fixed asymptotically. Also, suppose $C=0$. Then we have $\zeta_0 = O(n^{1+2/s'}R^2)$ with $R _ {\text{KAN}} =(\prod^{L} _ {j=1}\rho _ j)(\sum^{L}_{i=1}(B_ic_i)^{2/3})^{3/2}$, which measures the network complexity. It thus implies that the bound is of the order
> \begin{align*}
> O\left(\frac{R _ {\text{KAN}}}{n^{1/2-1/s'}}+\frac{1}{n^{(s-1)/(2s)}}\right).
> \end{align*}
> We note that the excess risk has an order close to the rate $O(n^{-1/2})$ when $s'$ and $s$ are large. In particular, if $G(\mathbf{x},y)$ and $B(y_i)$ both have sub-exponential tails (so that all their moments exist) and $R=O((\log(n))^a)$ for some $a>0$,  the excess risk has the rate $O(n^{-1/2})$ up to some logarithmic factors. See Section A.3 for the precise statements.
>
>
> **Interpretation of the rate in Corollary 2.** Next, we consider the upper bound on the excess risk in Corollary 2. The parameter $\xi_0$ has a similar interpretation as that of $\zeta_0$. It measures the model complexity and depends on the smoothness parameter $\nu$ of the Sobolev space, the network widths $d_i$, the underlying ranks $r_i$, the bounds on the RKHS norms of the activation functions $R_i$ and the Lipschitz constants $\rho_i.$ Again, ignoring the logarithmic terms, the bound is of the order
> \begin{align*}
> O\left(\frac{\xi_0^{\nu/\tilde{d}}}{n^{(\nu/\tilde{d}+1)/2}}+\frac{1}{n^{(s-1)/(2s)}}\right).
> \end{align*}
> In view of the definition of $\xi_0$, when $\tilde{d}>\nu$, we have
> \begin{align*}
> \xi_0^{\nu/\tilde{d}} = O\left(n^{2/s'+1/2}\right)
> \end{align*}
> and hence, the bound can be simplified to
> \begin{align*}
> O\left(\frac{1}{n^{\nu/(2\tilde{d})-2/s'}}+\frac{1}{n^{(s-1)/(2s)}}\right).
> \end{align*}
> When $G(\mathbf{x},y)$ and $B(y_i)$ both have sub-exponential tails (so that all their moments exist),  the excess risk has the rate $O(n^{-\nu/(2\tilde{d})})$.
> Note that when $\tilde{d}<\nu$, with similar arguments, the corresponding rate would become $O(n^{-1/2})$ up to some logarithmic factors.

---

> ### Author Response · Authors · 2024-11-25
> **Response to reviewer Reviewer vaj9: Part II**
>
> **Comparison with the results for MLP.** For a fair comparison, let us consider the bound derived in Theorem 2 with $C=0$ and $\max_i B(y_i)$ being finite because this setting is closest to the one for MLPs considered in [R1]. Recall from the paper that $\zeta = \tilde{\alpha}^3 \log(2\tilde{d}\tilde{p}) \max _ i B^2(y _ i)$. Ignoring the smaller order terms, the bound can be expressed as
>
> $$O\left(
> \frac{||\mathbf{X}||R _ {\text{KAN}}}{n} \log(n)\log(2\tilde{d}\tilde{p}) + \sqrt{\frac{\log(1/\epsilon)}{n}}\right),$$
>
> where $R _ {\text{KAN}} =(\prod^{L} _ {j=1}\rho _ j)(\sum^{L}_{i=1}(B_ic_i)^{2/3})^{3/2}$ is the complexity measure of KANs.
> There have been several results in the literature concerning risk bounds or estimation error bounds for MLP or feedforward networks, such as [R1, R3, R5]. The most closely related work is [R1], which shows that with probability greater than $1-\epsilon$, the error bound is of the order
>
> $$
> O\left(\frac{||\mathbf{X}||R _ {\mathcal{A}}}{\gamma n}\log(n)\log(W)+\sqrt{\frac{\log(1/\epsilon)}{n}}\right),
> $$
>
> where $R_{\mathcal{A}}$ is a measure of model complexity (similar to $R _ {\text{KAN}}$ defined above), $\gamma$ is the margin in the ramp loss and $W$ is the network width (similar to $\tilde{d}$ in our case).
>
> When $R$ and $R_{\mathcal{A}}$ are of the same order (denoted by $R\asymp R_{\mathcal{A}}$), the two bounds are also of the same order. This is not surprising, as $R \asymp R _ {\mathcal{A}}$ implies that the two classes have the same order of complexities.
> On the other hand, if $R = o(R _ {\mathcal{A}})$ and $R _ {\mathcal{A}}/\sqrt{n} \rightarrow +\infty$, then the bound for KANs is of a smaller order than the bound for MLPs. This situation occurs when the underlying true regression function can be more effectively approximated using KANs (e.g., the true regression function can be exactly expressed by a KAN network). As discussed in [R4], KANs may require fewer parameters than MLPs to approximate the same underlying true function because KANs leverage the inherently low-dimensional representation of the true function, whereas MLPs do not. Below, we provide a numerical example to support this claim.
>
> We use the symbol $\phi$ with different subscripts to denote different spline functions (linear combinations of spline basis). We consider the regression settings, $y=f(x)+\epsilon$, where $\epsilon$ is a mean-zero random error, and:
>
> **Example 1.** $f(x)=\sum_{j=0}^{99}\phi_j(x_j)$, which can be exactly represented by a KAN of shape [100, 1];
>
> **Example 2.** $f(x)=\phi_{1,0}(\sum_{j=0}^{99}\phi_{0,0,j}(x_j))+\phi_{1,1}(\sum_{j=0}^{99}\phi_{0,1,j}(x_j))$, which can be exactly represented by a KAN of shape [100, 2, 1].
>
> We employed KAN with shapes [100, 1], [100, 2, 1],  and MLP with shapes [100, 6, 1] and [100, 12, 1] for the two settings, respectively. Since the number of basis splines used in KAN is 6, the widths for the MLP were set so that the two networks have a similar number of parameters. The test errors (mean squared errors) are shown in the following table.
>
> |       |  Example 1   | Example 2 |
> |-------|---------|-----------|
> | KAN   | 2.0765  | 1.1184e-02|
> | MLP   | 10.5546 | 5.8772e-02|
>
> We observe that the test error of the MLP network is approximately five times that of the KAN network. We also tested MLPs with different shapes and observed that neither decreasing nor increasing the widths or depths significantly improved the test error. Although the examples considered are special cases where the regression functions can be exactly represented by a KAN, we point out that the MLP is a special structure of KAN (e.g., when the identity function is included as a basis function and ReLU is used as the activation function). Therefore, the KAN network has a greater capability to reflect the underlying true regression function, and when it does so, we can expect a lower test error. A similar phenomenon has been observed in [R4]; see Figures 3.1-3.3 therein.
>
> *3. It appears that Section 2.3 only considers RKHS induced by Matern kernel, instead of a general low-rank RKHS. Because of the equivalence between this specific RKHS and Sobolev space, would it be a better idea to change the title of Section 2.3 to ``Activation functions on Sobolev space''?*
>
> **Ans:** Thank you for the suggestion. In the revision, we have changed the title of Section 2.3 to ``Risk analysis: activation functions in low-rank Sobolev Space''.

---

> > ### Comment · Reviewer_vaj9 · 2024-11-28
> >
> > I sincerely thank the authors for thoroughly addressing my questions and significantly improving the paper's presentation. I am very pleased with the revisions, which have led me to increase my score to 8. In particular, the authors have clarified the implementation of the KAN architecture in the experiments and provided an in-depth discussion of the comparison between KAN and traditional MLP in terms of generalization. Overall, I believe this is a solid work.

---

### Author Response · Authors · 2024-11-24

First and foremost, we express our sincere gratitude to the five reviewers for their invaluable feedback, which has helped us improve the quality of our manuscript. We have revised our manuscript, taking all the reviewers' comments into consideration. Before providing point-by-point responses, we would like to summarize the major changes we have made to address the reviewers' main concerns.

1. We highlight our technical contributions relative to existing works and further discuss the implications of our main results.

2. We discuss the derived bounds and compare them with the corresponding bound for MLPs; see Section A.1 of the appendix.

3. We refine our excess risk bounds under the sub-exponential tail condition on $G(\mathbf{x},y)$ and $B(y)$; see Section A.2 of the appendix.

4. We derive a lower bound on the empirical Rademacher complexity for KANs; see Section A.3 of the appendix;

5. We provide additional numerical results and discussions in Section A.4 of the appendix.

We have highlighted the changes in red in the revision. Below, we provide point-by-point responses to each reviewer's comments.

References

[R1] Peter L Bartlett, Dylan J Foster, and Matus J Telgarsky. Spectrally-normalized margin bounds for neural networks. Advances in neural information processing systems, 30, 2017.

[R2] Alexander Dylan Bodner, Antonio Santiago Tepsich, Jack Natan Spolski, and Santiago Pourteau. Convolutional kolmogorov-arnold networks. arXiv preprint arXiv:2406.13155, 2024.

[R3] Max H Farrell, Tengyuan Liang, and Sanjog Misra. Deep neural networks for estimation and inference. Econometrica, 89(1):181–213, 2021.

[R4] Ziming Liu, Yixuan Wang, Sachin Vaidya, Fabian Ruehle, James Halverson, Marin Soljačić, Thomas Y Hou, and Max Tegmark. Kan: Kolmogorov-arnold networks. arXiv preprint arXiv:2404.19756, 2024.

[R5] Johannes Schmidt-Hieber. Nonparametric regression using deep neural networks with relu activation function. Annals of Statistics, pages 1875–1897, 2020.

[R6] Yixuan Wang, Jonathan W Siegel, Ziming Liu, and Thomas Y Hou. On the expressiveness and spectral bias of kans. arXiv preprint arXiv:2410.01803, 2024.

---

> ### Author Response · Authors · 2024-12-02
>
> We would like to thank all the reviewers for their efforts in the review process and helpful comments, which have improved the manuscript.

---

### Meta-Review · Area_Chair_qV88 · 2024-12-19

**Metareview:**

Summary:
This paper provides a theoretical analysis of Kolmogorov–Arnold Networks (KANs), including generalization bounds for regression tasks and analysis of network complexity. The work addresses both bounded and unbounded loss functions and offers empirical validation of theoretical insights. Despite some identified weaknesses, the contribution to understanding the generalization properties of KANs is solid, and the paper sets a valuable foundation for future work in this emerging area.

Strengths:
Novelty: The paper is among the first to establish theoretical bounds for KANs, contributing to the theoretical foundation of this novel architecture.

Thoroughness: The authors present comprehensive theoretical results, including complexity measures and empirical validation.

Relevance: Given the growing interest in KANs within the research community, the work provides timely insights into their properties.

Weaknesses:
Performance in Studied Regime: The empirical results indicate that KANs perform poorly in the examined regime, particularly without regularization.

Theoretical Impact: The derived bounds lack evidence of their ability to explain the practical performance of deep learning systems.
Lack of Insight on Advantages Over MLPs: The analysis does not provide a compelling explanation for why KANs might outperform MLPs in practice.

Innovation in Proof Techniques: While the theoretical results are sound, the proof techniques are incremental, building heavily on prior work in related areas.


Suggestions to Authors:

Acknowledge explicitly in the paper the identified weaknesses, including the limited empirical performance and lack of compelling theoretical justification for the superiority of KANs over MLPs.

Discuss the practical implications of the theoretical results more clearly, particularly in terms of how they could guide the design or regularization of KANs.

Consider adding further empirical experiments in future work, particularly in regimes where KANs are expected to perform competitively.

Conclusion:
Despite the highlighted weaknesses, the paper lays important groundwork for theoretical studies of KANs and will be a valuable resource for researchers interested in this architecture. I recommend acceptance, contingent on the authors incorporating suggested revisions to acknowledge and contextualize the limitations of their findings.

**Additional Comments On Reviewer Discussion:**

Discussion:
The discussion among reviewers identified the above weaknesses but converged on the view that the paper contributes a solid and well-reported theoretical foundation for KANs. The theoretical contributions, while incremental in some respects, are novel in their application to KANs and provide a framework for future investigations into their generalization properties.

The empirical results are a notable point of concern, as they do not showcase clear advantages of KANs in practical settings. However, the authors' analysis of network complexity and its relation to generalization offers valuable insights. This conceptual contribution may inspire further research into regularization techniques for KANs and broader investigations into their applicability.

---

### Decision · Program_Chairs · 2025-01-22

Accept (Poster)